# A Review of Approaches to Potentiate the Activity of Temozolomide against Glioblastoma to Overcome Resistance

**DOI:** 10.3390/ijms25063217

**Published:** 2024-03-12

**Authors:** Aniruddha S. Karve, Janki M. Desai, Sidharth N. Gadgil, Nimita Dave, Trisha M. Wise-Draper, Gary A. Gudelsky, Timothy N. Phoenix, Biplab DasGupta, Lalanthica Yogendran, Soma Sengupta, David R. Plas, Pankaj B. Desai

**Affiliations:** 1Division of Pharmaceutical Sciences, James L. Winkle College of Pharmacy, University of Cincinnati, Cincinnati, OH 45267, USA; karveas@ucmail.uc.edu (A.S.K.); gadgilsn@mail.uc.edu (S.N.G.);; 2Lapix Therapeutics, Cambridge, MA 02142, USA; 3Division of Hematology/Oncology, University of Cincinnati College of Medicine, Cincinnati, OH 45267, USA; 4Division of Oncology, Cincinnati Children’s Hospital, Cincinnati, OH 45229, USA; 5Department of Neurology & Rehabilitation Medicine, University of Cincinnati College of Medicine, Cincinnati, OH 45267, USA; yogendlv@ucmail.uc.edu; 6Department of Neurology, School of Medicine, University of North Carolina, Chapel Hill, NC 27514, USA; ssengup@email.unc.edu; 7Department of Neurosurgery, School of Medicine, University of North Carolina, Chapel Hill, NC 27514, USA; 8Department of Cancer Biology, University of Cincinnati College of Medicine, Cincinnati, OH 45267, USA

**Keywords:** glioblastoma, treatment resistance, combination therapeutics

## Abstract

A glioblastoma (GBM) is one of the most aggressive, infiltrative, and treatment-resistant malignancies of the central nervous system (CNS). The current standard of care for GBMs include maximally safe tumor resection, followed by concurrent adjuvant radiation treatment and chemotherapy with the DNA alkylating agent temozolomide (TMZ), which was approved by the FDA in 2005 based on a marginal increase (~2 months) in overall survival (OS) levels. This treatment approach, while initially successful in containing and treating GBM, almost invariably fails to prevent tumor recurrence. In addition to the limited therapeutic benefit, TMZ also causes debilitating adverse events (AEs) that significantly impact the quality of life of GBM patients. Some of the most common AEs include hematologic (e.g., thrombocytopenia, neutropenia, anemia) and non-hematologic (e.g., nausea, vomiting, constipation, dizziness) toxicities. Recurrent GBMs are often resistant to TMZ and other DNA-damaging agents. Thus, there is an urgent need to devise strategies to potentiate TMZ activity, to overcome drug resistance, and to reduce dose-dependent AEs. Here, we analyze major mechanisms of the TMZ resistance-mediated intracellular signaling activation of DNA repair pathways and the overexpression of drug transporters. We review some of the approaches investigated to counteract these mechanisms of resistance to TMZ, including the use of chemosensitizers and drug delivery strategies to enhance tumoral drug exposure.

## 1. Introduction

### 1.1. Identifying Molecular Markers for Glioblastoma

Glioblastoma (GBM), formerly known as glioblastoma multiforme, is the most malignant of adult gliomas, arising from glial cells or their progenitors. With a median survival of less than 15 months, GBMs have one of the worst prognoses. The latest edition of the WHO classification of CNS tumors (5th edition, 2021) [1] emphasizes the characterization of gliomas based on molecular attributes. GBMs were redefined according to molecular testing with respect to isocitrate dehydrogenase (IDH) status. Previously, GBMs had included high-grade IDH-wildtype and IDH-mutant neoplasms. However, with the revised taxonomy, the term GBM is now exclusively used to describe adult IDH-wildtype tumors.

Other pertinent markers include TP53 (tumor suppressor protein P53) mutations associated with the increased proliferation and invasiveness of GBM cells, B-RAF V600E, and GATA4 mutations, which contribute to enhanced chemoresistance. The overexpression of FGFR1 (fibroblast growth factor receptor 1) and loss of function PTEN (phosphatase and tensin homolog) mutations are associated with increased GBM proliferation [2]. Although these classifiers are somewhat useful for prognostic assessments, the striking inter-tumoral and intra-tumoral heterogeneity of mutations confound the use of these classifiers for the identification of optimal clinical treatment strategies.

Verhaak et al. (2010) suggested the classification of GBMs into proneural, neural, classical, and mesenchymal subtypes based on histology [3]. Wang et al. have helped in identifying a cluster of key transcriptional signatures and genetic mutations associated with these histological subtypes as shown in Table 1 [4]. Some publications suggest that neural subclassification is confounded by the presence of normal neural cells at the tumor margins. Due to this potential contamination of tumor cells with normal neural cells, the neural subtype is often not considered a distinct subtype [4,5,6].

#### 1.1.1. Epidemiology of GBMs

GBMs account for approximately 50% of all malignant brain tumors. According to the Central Brain Tumor Registry of the United States (CBTRUS), an estimated 12,910 new cases of GBMs were diagnosed annually between 2016–2020 in the US [7]. Unfortunately, there seems to be an increase in annual incidences, and in 2023, approximately 13,750 individuals were expected to be diagnosed with GBMs in the US. The global incidence of GBM was estimated to be 160,000 in 2020 [8]. The incidence of GBMs appears to increase with age, and the rates are highest in individuals in the age group ranging from 75 to 84 years [7]. The incidence rate for the diagnosis years between 2004 and 2016 exhibits gender-specific differences, as the incidence rate in males (3.99 cases every 100,000 adults) was significantly higher than that in females (2.52 cases every 100,000 adults) [9].

#### 1.1.2. Current Standard of Care for GBM

The current standard of care for GBM patients includes maximally safe surgical resection, followed by concomitant fractionated radiotherapy and oral chemotherapy with the DNA-methylating agent temozolomide (TMZ), taken 7 days a week for 6 weeks at the dose of 75 mg/m^2^. This is followed by six cycles of adjuvant TMZ administration at the dose of 150–200 mg/m^2^, with each 28-day cycle consisting of 5 days of TMZ administration [10,11,12]. Unfortunately, this multimodal approach is not curative, and only extends the median survival from 4 months to approximately 15 months [3]. FDA approval for TMZ was granted in 2005 based on the controlled open-label, multicenter trial conducted by Stupp et al., in which 573 GBM patients were randomized to receive either temozolomide and radiotherapy (*n* = 287) or radiotherapy alone (*n* = 286) [12]. The median OS was 14.6 months in patients receiving TMZ and radiotherapy versus 12.1 months (radiotherapy alone) [10,11]. The OS levels following treatment varies based on subtypes and the treatment modality, ranging from 10–23 months [3,4,5,6,7,8,9,12,13,14,15].

### 1.2. Pharmacology of TMZ

Following oral administration, the time (T_max_) required to achieve the maximal plasma concentration (C_max_) of TMZ ranges from 0.5 to 1.5 h, suggesting relatively rapid oral absorption [9,10]. The mean C_max_ of TMZ and MTIC at an oral dose of 150 mg/m^2^ are 7.5 µg/mL and 0.282 µg/mL, respectively. The mean area under the plasma–concentration time curve (AUC) of TMZ and MTIC after a single 90 min intravenous (IV) infusion of 150 mg/m^2^ of TMZ are 17.1 µg·h/mL and 1.0 µg·h/mL, respectively [16]. Some important clinical pharmacokinetic (PK) properties of TMZ are shown in Table 2. A comparison of the oral and IV C_max_ and AUC values suggests that the oral bioavailability of TMZ is close to 100%. The compound is more stable in acidic media (pH ≤ 5.0) than it is in basic media (pH ≥ 7.0) due to the protonated form that minimizes the catalytic process. As such, TMZ has high oral bioavailability, but a short half-life of about 2 h. The extent of plasma protein binding of TMZ is only 15%, and as such, a major fraction of the administered dose remains unbound in the systemic circulation.

TMZ has a small molecular weight (195 Da), and is also a relatively lipophilic compound (as shown in Figure 1). Other physicochemical properties of TMZ that conform to Lipinski’s rule of five are as follows: log *p* value < 5, number of rotatable bonds < 5, number of hydrogen bond donors and acceptors < 5, and a molecular weight < 500 Da [15]. These physicochemical attributes facilitate TMZ penetration across the blood–brain barrier (BBB). A number of pre-clinical and clinical studies have quantitated the blood–brain extracellular (ECF) partitioning of TMZ. Using microdialysis, we compared the plasma–brain ECF PK of TMZ in Sprague Dawley rats (Arora et al., 2018) [17]. Microdialysis is an invasive technique that makes use of a probe containing a semipermeable dialysis membrane in order to collect interstitial fluid from the target tissue. We determined the ratio of the peak concentration (C_max_) or AUC in ECF to the corresponding unbound plasma values. The partitioning of TMZ was approximately 33% and 45% on the bases of C_max_ and AUC, respectively. Similar studies by Zhou et al. (2007) reported a ratio of 43% based on a comparison of tumoral ECF C_max_ to plasma in xenografted athymic rats [18]. Brain interstitial microdialysis was also performed in a clinical setting by Portnow et al. (2009) [16]. The brain ECF C_max_ for TMZ at a single maintenance dose of 150 mg/m^2^ ranged from 1 to 8 µM [15]. Relative to the plasma drug levels, the TMZ brain partitioning ranged between 10–47%. The concentrations of TMZ needed to induce 50% cytotoxicity (IC_50_) in drug-sensitive GBM cells are in the range of 1–5 µM [19,20]. However, when the tumor cells develop drug resistance, the IC_50_ values often exceed 100 µM [19,20]. There is little accumulation at the steady state following multiple dosing, given the short elimination half-life of TMZ [16,17,18]. Additionally, clinically relevant doses of TMZ can be associated with common grade 3–4 AEs, including neutropenia (5–8%), thrombocytopenia (4–11%), fatigue (9%), and headaches (4%). Clearly, with the development of resistance, the tumoral concentrations of the drug become sub-therapeutic, leading to tumor recurrence, and, with the dose-limiting AEs, there are limited alternatives to deploy effective therapeutic agents for recurrent GBMs [11,12,13].

## 2. FDA-Approved TMZ Combination Approaches for GBM Treatment

As shown in Table 3 and as described below, TMZ is often combined with other therapeutic approaches approved for GBM.

**TMZ in combination with Gliadel^®^ wafers**: Combination therapy with TMZ and alkylating agent BCNU (1,3-bis(2-chloroethyl)-1-nitroso-urea), approved in 1977 as an injectable chemotherapy for GBM, has been employed with limited success. The systemic administration of this drug is associated with serious adverse events (AEs), such as myelosuppression and pulmonary toxicities, which limit its clinical use as an injectable therapy [21,22]. An alternative approach is the localized delivery of BCNU as Gliadel^®^ wafers, which are implanted at the time of tumor resection. Each wafer contains 192.3 mg biodegradable copolymer polifeprosan 20 poly [bis(p-carboxyphenoxy)] propane and sebacic acid (in a 20:80 molar ratio) and 7.7 mg BCNU. Despite the theoretical rationale, this combination therapy approach does not significantly improve the OS or progression-free survival (PFS) levels. Moreover, the use of Gliadel^®^ therapy is associated with severe CNS toxicities, including seizures, intracranial hypertension, and impaired wound healing. The risk of wafer migration and meningitis have also been associated with this approach [21,22].

**TMZ in combination with anti-angiogenic compounds**: Since GBMs are highly vascularized and abundantly express a vascular endothelial growth factor (VEGF) that promotes angiogenesis, numerous anti-angiogenic agents have been investigated as potential treatments for newly diagnosed and recurrent GBMs. Most prominent among these is bevacizumab (BVZ), a targeted therapeutic antibody that binds and inhibits VEGF. Marketed as Avastin^®^, BVZ was approved by the FDA to be used as a treatment for recurrent GBMs in 2009 [23,24]. Although BVZ is well tolerated, an impaired VEGF function is linked to multiple toxicities, including hypertension, thromboembolic events, gastrointestinal perforation, cerebral bleeding, wound-healing complications, and proteinuria [23]. Gilbert et al. (2014) compared BVZ and TMZ versus TMZ alone for newly diagnosed GBM patients. While the PFS rate increased from 7.3 to 10.7 months, the impact on the OS rate was minimal (15.7 vs. 16.1 months). Combined treatment also resulted in increased side effects and a higher symptom burden, as well as decreased neurocognitive function with a worsening QoL [23]. Similar results were reported by Chinot et al. (2014) [24]. Overall, a comprehensive analysis of 11 randomized controlled trials of various anti-angiogenic therapies against high-grade gliomas by Ameratunga et al. (2018) concluded that the combination of anti-angiogenic drugs with chemotherapy may lead to a marginal increase in PFS rate for both newly diagnosed and recurrent GBM patients, but has minimal impact on OS rate or the QoL [25].

**TMZ in combination with TTFields**: Another novel therapeutic approach entails the use of tumor treating fields (TTFields), which are intermediate-frequency (~100–500 kHz) and low-intensity (1–3 V/cm) electromagnetic fields generated through a portable device containing transducer arrays, placed directly on the skin surrounding the tumor location. TTFields have been approved as a medical device (Optune^®^) to be used in conjunction with TMZ for the treatment of adult patients with newly diagnosed, supratentorial GBMs following surgery and radiation therapy [26]. In a Phase III clinical trial, patients with newly diagnosed GBMs were administered TTFields (200 kHz) in combination with TMZ and had a median PFS rate of 6.5 months, a significant improvement relative to 3.9 months in patients treated with TMZ alone. Likewise, the OS rate increased to 17.4 months for patients receiving TTF with TMZ vs. 13.7 months for patients treated with TMZ alone. The systemic AEs associated with this combination are similar to TMZ alone, so it appears that TTFields do not carry a risk for increased systemic AEs [27]. However, TTFields cause local irritation or allergic contact dermatitis at the site of the transducer array attachment, resulting from prolonged exposure to sweat, hydrogels, adhesives, and/or a combination of these factors. These local complications often require topical corticosteroids, the modification of array positioning, and/or protecting the skin with sterile dressing pads. Overall, TTFields therapy is patient-friendly, non-invasive, portable, and appears to produce a modest improvement in the OS rate and the QoL of GBM patients. However, a major limitation of TTFields is that it does not seem to be effective in the recurrent setting [27].
ijms-25-03217-t003_Table 3Table 3FDA-approved combination of TMZ for GBM therapy.Combination TreatmentOutcome MeasuresCurrent Status as GBM Therapeutic ReferencesTMZ + Gliadel^®^No significant impact on OS or PFSPossible off target effectsFDA-approved for newly diagnosed GBM[22]TMZ + BevacizumabIncrease in PFS by ~3 monthsNo significant impact on OS (<1 month increase) when compared with current standard of careFDA-approved for newly diagnosed GBMs [23,24,25]TMZ + TTFieldsIncrease in OS (~4 months) and PFS (~3 months)Skin toxicities at the site of application of TTFields deviseFDA-approved combination for newly diagnosed GBMs following tumoral resection and the completion of radiotherapy Phase III trials in combination with radiotherapy[27]TMZ + Pembrolizumab + TTFieldsIncrease in OS (~10 months) and PFS (~6 months)Phase II trials[28]

## 3. Development of Resistance to TMZ

Resistance to TMZ and other DNA-alkylating agents is a major barrier to the successful treatment of GBM. While GBM tumor cells may be intrinsically resistant to TMZ, they more frequently acquire resistance to TMZ after the initiation of therapy. As indicated above, with the development of drug resistance, the IC_50_ values for TMZ often exceed 100 µM, which are clinically unattainable drug concentrations. The dose-dependent AEs associated with TMZ restrict the dose and dosing frequency to combat non-responsive GBM. Furthermore, due to its short elimination half-life of 1.8–2 h, systemic and brain/tumoral levels cannot be sustained for the period necessary for therapeutic action, and the dose-dependent AEs of TMZ prohibit the use of higher doses [10,11,12,27,29]. The mechanisms of resistance to TMZ elaborated below are multifarious. Some of the principal mechanisms include the following: (1) the hyperactivation of DNA repair mechanisms, (2) the acquisition of the glioma stem cell phenotype, (3) the increased expression of drug efflux transporters, (4) mutations in signaling pathways affecting DNA-damage repair, and (5) GBM epigenetics modulated by histone deacetylases (HDACs) and microRNAs.

### 3.1. Hyperactivation of DNA Repair Mechanisms

The mechanism of TMZ entails the DNA-methylation of guanine at the O^6^ position (O^6^-meG), as well as 7-methylguanine (N^7^-meG) and 3-methyladenine (N^3^-meA). While N^7^-meG and N^3^-meA account for 70% and 9% of the adducts formed, respectively, these are efficiently repaired by the base excision repair (BER) pathway, and they normally contribute little to cytotoxicity. The O^6^-meG accounts for only 6% of the adducts formed, but represents the primary mediator of TMZ cytotoxicity [19,30].

**MGMT-mediated DNA repair:** MGMT is an enzyme which is responsible for the repair of DNA damage resulting from the methylation of O^6^-guanine (O^6^-meG), which is formed by TMZ. It removes the O^6^-alkylguanine DNA adduct through the covalent transfer of the alkyl group to the conserved active-site cysteine, thus restoring the guanine to its normal form and evading DNA strand breaks (Figure 1). MGMT acts both as a transferase and as an acceptor of the alkyl group. After receiving a methyl group from O^6^-meG, MGMT is inactivated by ubiquitin-mediated degradation, and thus it is a suicidal protein. The presence of MGMT degrades the O6-meG produced by TMZ and protects against cell death. MGMT is ubiquitously expressed in normal human tissues, but is overexpressed in many human tumors, including lung cancer, colon cancer, breast cancer, and GBMs [19,30].

MGMT promoter demethylation is the key mechanism that regulates MGMT gene expression and predicts TMZ response in GBM patients. In many primary GBM tumors, the MGMT promoter is in its methylated (repressed) state, and thus is expressed at relatively low levels, rendering cells vulnerable to TMZ-induced cell death. However, in many cases, especially in recurrent GBM, the MGMT promoter is in its unmethylated (active) state, which leads to the repair of methylated nucleotides and the development of resistance to TMZ treatment (Kitange et al., 2009) [19]. MGMT overexpression also renders cells resistant to other DNA-alkylating agents, limiting the use of many other chemotherapeutic compounds alongside TMZ [19].

**Mismatch repair:** If an O^6^-meG DNA adduct escapes MGMT repair, it forms a base pair with thymine during DNA replication. The futile cycling of this mismatched base pair (O^6^-meG:T) by the mismatch repair (MMR) pathway removes the thymine, but leaves the methylated guanine, resulting in DNA double-stranded breaks, irreparable genomic damage, and the activation of apoptotic cell death [31]. The recognition of TMZ-induced O^6^meG:T mismatches is initiated by the multiprotein complex MutSα (comprised of the heterodimer of MSH2 and MSH6). In the absence of a functional MMR response, O^6^meG:T is not targeted for this “futile repair” cycle, and the cells avoid apoptosis and survive. Several studies suggest that MSH2 and MSH6 inactivation may also contribute to the development of MMR deficiency, leading to cell survival and the emergence of resistance to TMZ [31,32]. McFaline-Figueroa et al. (2015) have shown that even a modest decrease in MSH2 and MSH6 expression leads to TMZ resistance in GBM cells [33]. Collectively, these studies underscore the substantial reduction in TMZ activity due to MSH2 and MSH6 attenuation, and underscore that MMR activity offers a predictive marker for therapeutic responses to TMZ.

**Base excision repair:** Although N^7^-methylguanine and N^3^-methyladenine represent the major TMZ-induced methylation products, these adducts are readily repaired by the DNA glycosylase class of enzymes from the BER pathway. These enzymes mediate the removal of methylated purine adducts, which is followed by the repair of the apurinic (AP) site from the DNA with AP endonuclease 1, DNA ligase, and DNA polymerase. The overexpression of DNA glycosylases in GBM tumors is often associated with resistance to TMZ and poor survival [34].

In that regard, another class of enzymes, poly-(ADP-ribose) polymerases (PARP), particularly PARP1, is also involved in the repair of SSBs induced by DNA alkylation through the BER pathway. PARP1 also facilitates the recruitment of MGMT proteins for the repair of O^6^-methylguanine residues, and, hence, the overexpression of PARP1 and the decreased PARP1 cleavage are commonly associated with increased chemoresistance and poor survival in GBM patients [35]. The small molecule inhibitors of PARP1 provide an attractive strategy to inhibit the BER and MGMT pathway-mediated DNA-damage repair in GBM. [36,37].

### 3.2. Glioma Stem Cells (GSCs)

GSCs constitute a subpopulation of GBM cells that display high levels of cellular plasticity and tumor-initiating capabilities. With functional characteristics that permit sustained self-renewal, persistent proliferation, and tumor initiation, GSCs play a key role in the resistance of these tumors to conventional therapies, as well as recurrent disease. GSCs are the key reason for the high degree of heterogeneity in GBM cells, which can lead to the propagation of advantageous mutations, in turn leading to resistance to chemo- and radiotherapy. While some of the molecular markers for GSCs have been identified as CD133, ALDH1A3, SOX-2, and Nestin, these cells are continuously evolving on the stemness–differentiation axis, and it is challenging to identify and therapeutically target the entire subpopulation of GSCs in the tumor mass [38]. Furthermore, GSCs often exhibit mutations in DNA-damage repair mechanisms, such as unmethylated MGMT [19] and MSH2/MSH6 deactivation [31], mediated through alterations in intracellular signaling pathways like the Wnt/β-catenin pathway [39], the JAK/STAT pathway [37], and EGFR signaling [38], thus leading to resistance to conventional chemotherapy with TMZ. Targeting these GSC-specific signaling transduction pathways represents a major focus for chemosensitization to TMZ [38].

### 3.3. Barriers to Brain/Brain Tumor Permeability: Drug Efflux Transporters

A formidable barrier that impacts the development and rational use of any drug targeting the CNS is the BBB. Many otherwise-effective chemotherapeutic agents are unable to cross the BBB, resulting in sub-therapeutic exposure. The CNS is separated from the blood via a vasculature, comprised of highly specialized endothelial cells that form part of the BBB. The unique physiologic characteristics of the BBB restrict the passage of most xenobiotics and high-molecular weight endogenous substances into the brain parenchyma. Brain endothelial cells differ markedly from those in other organs due to the fenestrated capillaries and intercellular clefts found in the capillary beds. Thus, in addition to passive diffusion and the facilitated uptake process of transcellular movement, these pores/fenestrae allow the entry of a diverse array of molecules, including polar/water-soluble compounds [40]. In contrast, brain endothelial cells have continuous tight junctions, an absence of fenestrations, and low pinocytic activity. Additionally, many efflux proteins belonging to the ATP-binding cassette (ABC) transporter family such as ABCB1 (P-glycoprotein; P-gp), ABCC1 (MRP1), ABCC2 (MRP2), ABCC4 (MRP4), and ABCG2 (breast cancer resistance protein; BCRP) expressed in the endothelial cells participate in active efflux of drugs to the blood. The presence of these multi-drug efflux transporters further restricts the entrance of xenobiotics and creates a “pharmacological sanctuary” [40,41,42]. Permeability is also impacted by the astrocytic and pericytic foot processes covering most of the cell surface and biochemical factors released by astroglia. The BBB represents the tightest endothelial barrier within the cardiovascular system, characterized by very low ionic permeability. A systematic measurement of trans-endothelial electrical resistance, reported by Butt et al. in 1990, was the first quantitative assessment of high electrical resistance in brain capillaries [43]. In general, unless a molecule is small, nonpolar, or a gas, crossing the BBB usually requires a dedicated transporter. Hence, drugs can only enter the brain primarily via transcellular passive diffusion, and, in some cases, via active uptake across the BBB. Accordingly, only small molecular weight compounds that are unionized at the physiological pH, and which are not substrates for efflux transport by the ABC transporter family, are able to penetrate the BBB at therapeutically relevant concentrations [43,44].

There is some evidence that the above-described tight junction structure may be compromised in primary and metastatic brain tumors due to increases in the perivascular space and the disruption of the BBB. As such, it appears that the blood–tumor barrier (BTB) may be less restrictive than the BBB. Nonetheless, even with such tumors, complexity is imposed by the overexpression of efflux transporters at the BTB that may compensate for increased passive diffusion. The result is that, in general, most therapeutic agents do not cross the BBB at clinically relevant levels [44]. TMZ is a substrate of both P-gp and BCRP, and the expression of these transporters is often elevated as a consequence of treatment with chemotherapeutic agents. The brain penetration of TMZ is 1.5-fold higher in P-gp and BCRP knockout mice, relative to wild-type mice [45]. Additional support for TMZ being a substrate for these efflux transporters is derived from studies indicating that the deletion of genes encoding P-gp and BCRP, or the pharmacological inhibition of these proteins with elacridar, significantly enhanced the anti-tumor efficacy of TMZ against orthotopically implanted GBM tumors in mice [45]. Thus, the evidence suggests that the elevated expression of the ABC transporters also contributes to acquired drug resistance, and these efflux transporters may be putative drug targets to circumvent TMZ resistance.

### 3.4. Mutations in Signaling Pathways Affecting DNA-Damage Repair

Extensive efforts have been made to discern the contribution of altered signaling pathways that reduce DNA damage mediated by anti-cancer drugs. Some of the major pathways that apparently contribute to TMZ resistance are briefly discussed below.

#### 3.4.1. Epidermal Growth Factor Receptor (EGFR) Variants

EGFR variants that impact chemotherapy are observed in more than 60% of GBM. These include EGFR amplification, the abnormal expression of EGFR agonists, the abnormal proteolytic release of EGFR agonists, and the intracellular activation of EGFR without the extracellular agonist binding constitutive amplification of EGFRvIII, the most common mutant, is present in approximately 20% of GBMs. Amplification in EGFRvIII expression leads to the activation of cell survival pathways, such as PI3K/Akt and MAPK/Erk, which rescue cells from the DNA damage induced by TMZ [46,47].

#### 3.4.2. Wnt Signaling Pathways

Mutations in canonical (β-catenin-dependent) and non-canonical (β-catenin-independent) Wnt signaling pathways have also been shown to enhance chemoresistance through the upregulation of MGMT and the promotion of tumor cell proliferation in GBMs [48]. Amplification in the production of Wnt3a, a ligand in the canonical Wnt/β-catenin pathway, is commonly observed in GBMs. Wnt3a induces stemness in glioma cells, leading to chemoresistance. The non-canonical Wnt pathway enhances the tumorigenicity and proliferation of GBM cells, and leads to the induction of NF-κB, in turn, promoting tumor cell survival. Both the canonical and non-canonical Wnt signaling pathways are putative targets for the treatment of recurrent GBMs and for the potentiation of TMZ activity in such tumors [48,49].

### 3.5. GBM Epigenetics and MicroRNAs

Epigenetic modification, such as the modulation of histone acetylation through histone deacetylases (HDACs), also appears to be an important mechanism of chemoresistance in GBM. The upregulation of several HDACs correlates to tumor grade, classification, and progression. Kitange et al. (2012) described the HDAC-mediated upregulation of MGMT in patient-derived GBM cells [50]. Hanisch et al. (2022) further detailed that class I HDACs (HDAC1/3/8) stimulate the E3 ubiquitin ligase RAD18, an enzyme involved in MGMT-mediated DNA-damage repair in GBMs [51]. Furthermore, both studies demonstrated that the pharmacological inhibition of these HDACs through small molecule inhibitors or downregulation using siRNAs sensitize GBM cells to treatment with TMZ [50,51].

MicroRNAs (miRNAs) are non-coding RNAs that regulate post-transcriptional gene expression. As described by Shi et al. (2010), miR-21, an oncogenic miRNA, is often upregulated in GBMs, resulting in the downregulation of tumor suppressor genes such as Bax [52]. The amplification of miR-21 results in reduced caspase-3 activity and inhibits GBM apoptosis induced by TMZ [52]. Furthermore, Zhang et al. (2012) demonstrated that the downregulation of miR-21 increases caspase-3 activity in U251 GBM cells and enhances TMZ-mediated apoptosis in these cells [53].

## 4. Approaches for Overcoming TMZ Resistance in GBM

The two primary approaches for improving therapeutic outcomes of TMZ therapy include (1) identifying/developing chemosensitizers that have the potential to circumvent the development of TMZ resistance or synergistically enhance its activity in drug-resistant tumors, and (2) developing novel formulations that enhance systemic and intra-tumoral drug persistence.

Table 4 lists some of the agents investigated as chemosensitizers to potentiate TMZ activity in drug-resistant GBM cells, and the mechanistic bases for their use are indicated below.

### 4.1. DNA-Damage Repair-Targeting Drugs

Given the above-indicated mechanistic bases, comprehensive attempts have been made to identify compounds that target MGMT, BER, and MMR pathways for the chemosensitization of TMZ-resistant GBM cells. As indicated previously, PARP is an enzyme that is required for MGMT activation and the repair of TMZ-induced DNA damage. PARylation, the process of the post-translational modification of MGMT, is necessary for the MGMT-mediated removal of O^6^-methylguanine adducts. In this regard, studies have suggested that PARP inhibitors (PARPi), such as veliparib and olaparib, restore chemosensitivity to TMZ in MSH6-inactivated, MMR-deficient GBM cells. These results identified a genetically defined subgroup of recurrent gliomas that may benefit from the combinatorial therapy of TMZ and PARPi [54]. Several PARPis, including olaparib, veliparib, niraparib, and pamiparib, are under clinical investigation in combination with TMZ in GBM patients. A potential advantage of using PARPis is that some of the clinically used compounds have the physicochemical properties needed to effectively penetrate the BBB and BTB [54,55].

Another approach entails the use of MGMT-depleting drugs. In that regard, bortezomib, a drug approved for the treatment of multiple myeloma, depletes MGMT levels in patient-derived unmethylated GBM cells, resulting in the re-sensitization of TMZ-resistant unmethylated GBM cells [56]. The combination of bortezomib and TMZ is currently being investigated in a multicenter, open-label, single-arm, non-randomized Phase IB/II trial (BORTEM-17) in recurrent GBM patients. Bortezomib (1.3 mg/m^2^) is administered intravenously on days 1, 4, and 7 during a 4-week cycle, and TMZ 200 mg/m^2^ is administered on days 5–7 [NCT03643549]. The promising interim analysis of this ongoing trial indicates that the combination treatment prolonged the median OS when compared with the standard of care treatment (19.0 vs. 12.2 months) in MGMT-unmethylated age-matched patients [57].

Examples of other compounds that target the BER pathway include methoxyamine, which blocks the AP sites and inhibits DNA repair. This compound was shown to significantly enhance the cytotoxicity of TMZ independently of MGMT status in the LN-428 GBM cells line [34]. Recent studies have utilized siRNA therapies targeted to BER proteins to enhance the sensitivity of resistant GBM cells to TMZ. Efforts have also included the development of formulations for the codelivery of the siRNA-targeting BER pathway and TMZ, which is being investigated in early-stage clinical trials [34].

### 4.2. Estrogen Receptor Modulators

Estrogen receptor alpha (ERα) and estrogen receptor beta (ERβ) have contrasting roles in glioma cells. ERα activation through estradiol leads to the activation of ERK/MAPK and PI3K/Akt signaling pathways, which promote cell proliferation and prevent the induction of apoptosis in GBMs [58]. With this rationale, the ERα antagonist tamoxifen has been assessed for its anti-GBM activity in the clinical setting. The results have been mixed, and most studies have required high doses of tamoxifen, perhaps due to limited BBB permeability [59]. For breast cancer therapy, tamoxifen is typically used at doses of 20 mg (b.i.d.). A Phase II clinical study by Spence et al. (2004) assessed TMZ and tamoxifen in recurrent astrocytic gliomas. The tamoxifen dosing scheme was 40 mg b.i.d. for 1 week, and was then increased in three successive weeks to 60, 80, and then 100 mg b.i.d. Significant toxicities (transaminitis, pancytopenia, 1st division herpes zoster, deep vein thrombosis, and fatigue) were observed, and the trial was halted [60]. In a subsequent study, Di Cristofori et al. (2013) investigated continuous tamoxifen (80 mg/m^2^ daily) with a dose dense TMZ (75–150 mg/m^2^ one week on/one week off) regimen in recurrent GBM patients. The median OS rate and time to tumor progression after recurrence were 17.5 and 7 months, respectively. The high-dose tamoxifen was well tolerated in this group of patients [61]. Some of the debilitating side effects of the drug include thrombocytopenia, hot flashes, and fatigue [62]. In another Phase I study, Patel et al. (2012) determined 100 mg/m^2^ as the maximal tolerated dose when given concurrently with temozolomide 75 mg/m^2^ and radiation therapy in high grade glioma patients (grade 3; *n* = 2 and GBM; *n* = 15) [63]. Overall, the use of ERα antagonists to potentiate TMZ activity appears to be a promising approach, but a safer alternative to tamoxifen or the optimization of its dosing regimen is warranted.

ERβ activation leads to the rapid phosphorylation of p38/MAPK and the induction of apoptosis through capase-3 activation and PARP cleavage. Sareddy et al. (2016) described the role of ERβ activation in the downregulation of DNA-damage repair in GBMs and the enhancement of chemotherapeutic activity of TMZ. They observed an increase in DNA damage in cells overexpressing ERβ following treatment with TMZ. Furthermore, a selective ERβ agonist, erteberel, sensitized GBM cells to several DNA-damaging drugs such as cisplatin, lomustine, and TMZ in vitro, significantly reducing the growth of orthotopically implanted GL26 GBM cells in mice [64].

### 4.3. Aromatase Inhibitors

Aromatase, a cytochrome P450 enzyme encoded by the gene CYP19A1, is overexpressed in GBM. Aromatase catalyzes the final step of estradiol synthesis and has been targeted by aromatase inhibitors for the treatment of estrogen receptor-positive breast cancers [65]. Our studies have shown that letrozole at non-cytotoxic concentrations causes a marked reduction in the IC_50_ values of TMZ against TMZ-sensitive and TMZ-resistant patient-derived GBM cells, through the significant enhancement of DNA damage and apoptosis imparted by TMZ [20]. Letrozole effectively penetrates the BBB and localizes in the tumoral region in orthotopic GBM tumor-bearing rats, and also has no pharmacokinetic drug–drug interactions with TMZ [17,66,67]. Thus, the use of aromatase inhibitors in combination with TMZ could prove to be a novel therapeutic approach for the treatment of GBM, and significant efforts to repurpose such FDA-approved drugs are currently in progress.

### 4.4. EGFR Inhibitors

Given the prevalence of EGFR mutations in GBM cells and the resulting impact on TMZ activity, the inhibition of the EGFR signaling pathway is a putative target for the sensitization of TMZ-resistant GBMs [46,47,68,69]. In fact, several compounds such as gefitinib and erlotinib have been investigated in the clinical setting for GBM therapy. However, many of these first-generation EGFR inhibitors such as gefitinib and erlotinib are hampered by poor BBB permeability. A newer EGFR inhibitor, osimertinib, appears to be a BBB penetrant. In a pre-clinical study, Chagoya et al. (2020) observed that this compound was quite effective against EGFRvIII-positive GBMs in vitro and in mice bearing orthotopically implanted GBMs [46]. Furthermore, Cardona et al. assessed the efficacy of the combinatorial treatment of GBMs with osimertinib and bevacizumab as a second line of therapy after the current standard of care treatment, including radiation therapy and adjuvant TMZ. The combination was found to be only marginally effective, most likely due to secondary alterations in GBM genetics, such as MET amplification and alterations in PDGFR, PTEN, and/or STAT3 [47]. Currently, other EGFR inhibitors, such as lorlatinib, are being assessed for safety and efficacy in combination with TMZ for the treatment of metastatic brain tumors [70]. Perhaps these observations will facilitate designing a safer and potentially more effective dosing regimen for other EGFR inhibitors for clinical investigations in GBM patients [69,70].

### 4.5. Wnt/β-Catenin Pathway Inhibitors

The Wnt/β-catenin pathway induces chemoresistance in GBMs through multiple pathways. First, as shown by Wickström et al. (2015), this pathway is involved in the upregulation of MGMT gene expression. The authors observed that the pharmacological and genetic inhibition of Wnt downregulated MGMT expression restored sensitivity to DNA-alkylating agents in GBM mouse models [48]. Several Wnt signaling inhibitors, including celecoxib, salinomycin, Wnt-C59, and LGK974, enhanced TMZ-induced cytotoxicity in colon cancer, medulloblastoma, and glioma cell lines [48]. Second, the Wnt/β-catenin pathway is instrumental in the development of GSCs. As shown by Behrooz et al. (2021), there is significant crosstalk between CD133, telomerase, and canonical Wnt pathway ligands. In addition to this role of canonical Wnt, this study also highlighted the upregulation of cellular Myc (c-Myc) and cyclin-D1 through β-catenin, and its contribution to the development of GSCs [39]. Third, the Wnt/β-catenin pathway also contributes to the increased expression of P-gp and BCRP in various cancers, which potentially enhances TMZ efflux and restricts its entry within the brain tumor mass [39]. Shen et al. (2013) demonstrated that the inhibition of Wnt/β-catenin signaling leads to the downregulation of P-gp and decreases the efflux of P-gp substrates in cholangiocarcinoma [49]. In another study, Laksitorini et al. (2019) demonstrated that the inhibition of β-catenin binding to transcription factor TCF-4 by small-molecule inhibitor ICRT-3 leads to a significant reduction in the levels of P-gp and BCRP in human cerebral microendothelial cells (hCMEC/D3) [71]. Thus, theoretically, Wnt/β-catenin inhibitors may enhance TMZ activity in resistant GBMs through multiple mechanisms that entail the inhibition of signaling pathways involved in MGMT regulation, the development of GSCs, and the expression of efflux transporters, P-gp, and BCRP.

### 4.6. Histone Deacetylase (HDAC) Inhibitors

HDACs have also been implicated in the upregulation of the signaling pathways responsible for the development of resistance to chemotherapeutics in GBMs. In particular, HDAC1, HDAC6, HDAC8, and HDAC11 are among the members of this enzyme family that have been most commonly associated with chemoresistance in GBMs. Pan-HDAC inhibitors are currently being assessed in pre-clinical and clinical settings as GBM therapeutics in combination with TMZ [50,72]. Ongoing clinical studies involving the combination of HDAC inhibitors and TMZ therapy are listed in Table 4. For example, Gatti et al. (2014) used vorinostat, a pan-HDAC inhibitor, in combination with TMZ for the treatment of BRAF-mutant melanoma. Combining this agent with TMZ resulted in a marked increase in efficacy against GBM lines in vitro and in orthotopic tumor-bearing mice [72]. Furthermore, Guntner et al. (2020) demonstrated that vorinostat effectively penetrates the BBB in pediatric CNS tumors [73]. A Phase II clinical trial to evaluate the efficacy of vorinostat when given in combination with TMZ and radiation in newly diagnosed GBM patients is in progress [Clinical trial identifier: NCT01236560].

### 4.7. Cell Cycle Checkpoint Inhibitors

Cyclin-D1, a pro-survival protein, is upregulated in GBMs through various intracellular pathways, including the Wnt/β-catenin pathway. The upregulation of the cyclin-D1 and cyclin-D1-CDK4 complex leads to the inhibition of the retinoblastoma protein (Rb), which then results in increased cell cycle progression from the G1 phase to the S phase. Inhibitors of cyclin-dependent kinase 4/6 (CDK4/6) have been shown to halt the cell cycle in the G1 phase and inhibit tumor proliferation [74,75]. Ribociclib, palbociclib, and abemaciclib are among the CDK4/6 inhibitors that have been evaluated as potential therapeutics for recurrent GBMs [74,75]. Only abemaciclib was effective in increasing the PFS in recurrent GBM patients in a Phase II clinical trial, although it did not alter the OS rate [74]. Tien et al. observed that, while ribociclib has good CNS penetration in recurrent GBM patients in a Phase 0 trial, the compound was not effective as a monotherapy [76,77]. Likewise, palbociclib was also found to be ineffective as a monotherapy in recurrent GBMs [76]. These observations suggest that CDK4/6 inhibitors may be a valuable addition to the armamentarium of drugs for the treatment of GBM, but a combination of these agents with other cytotoxic drugs may be critical for improving OS rates beyond the current standard of care [75,76,77].
ijms-25-03217-t004_Table 4Table 4Investigational status of selected compounds assessed for the potentiation of TMZ activity against GBM. (For BBB permeability, ‘+’ indicates published evidence of BBB permeability; ‘-’ indicates limited or no published data to ascertain BBB permeability).Drug ClassTMZ-Resistance Pathway TargetedExamplesCurrent Status BBB PermeabilityReferences/Clinical Trial IdentifierPARP inhibitorsDNA-damage repair(MGMT, BER);GSCsOlaparibPhase II trials+[54]VeliparibPhase II trials+[55]NiraparibPhase 0 trials+NCT05076513Proteasome inhibitorsDNA-damage repair(MGMT)BortezomibPhase Ib/II trials-NCT03643549Estrogen receptor modulatorsDNA-damage repairTamoxifenPhase II trials+NCT04765098ErteberelPre-clinical+[64]Aromatase inhibitorsDNA-damage repairLetrozolePhase 0/I trials+NCT03122197TKIsEGFR; MGMT expressionGefitinibPhase II trials-[69]OsimertinibPhase II trials+[47]LorlatinibPhase II trials-[70]Wnt signaling inhibitorsMGMT expression;P-gp expression;GSCsCelecoxibPhase II +NCT00047294SalinomycinPre-clinical-[46]HDAC inhibitorsHDAC1, HDAC6, HDAC8, HDAC11VorinostatPhase II trials+NCT01236560

PanabinostatPhase II trials+[73]CDK4/6 inhibitorsCyclin-D1; Rb1; Cell proliferationRibociclibPhase II trials+NCT05843253PalbociclibPhase II trials-[76]AbemaciclibPhase II trials+NCT02981940


### 4.8. Inhibitors of Multidrug Transporting Proteins

Approaches to inhibit the activity of these efflux transporters by competitive inhibitors or by altering the signaling pathways that participate in the regulation of these drug efflux proteins, such as Wnt/β-catenin, have been investigated for potentiating TMZ activity in GBM. For instance, well-known inhibitors of P-gp and BCRP, such as elacridar (GF120918), enhanced the brain penetration of TMZ 1.5-fold and increased its anti-tumor efficacy in mice with intracranial tumor implantations [45]. Reversan, a pyrazolopyrimidine, identified from a systematic screening of a compound library, is another inhibitor of efflux transporters that markedly enhanced TMZ activity in patient-derived primary and recurrent GBM lines [78]. Another approach entails targeting carbonic anhydrase II (CAXII), since this membrane-bound enzyme maintains intracellular/extracellular pH for efficient P-gp activity. Salaroglio et al. (2018) suggest that CAXII and P-gp are co-expressed in GBM lines and that a CAXII inhibitor, Psammaplin C, reduced P-gp-mediated TMZ efflux and potentiated its activity in GBM neurospheres. This effect was observed in GBM lines that had fully methylated, fully unmethylated, and partially methylated GBM lines [79]. These observations suggest that enhancing intracellular TMZ concentrations by the inhibition of efflux transporters may potentially re-sensitize GBM cells to TMZ. Despite the promising pre-clinical support, the approach of inhibiting P-gp and/or other efflux transporters for reversing resistance to cancer drugs has not been clinically successful thus far. The complexity of the resistance mechanisms and the safety and tolerability of reversing agents at the doses needed may be the limiting factors. This still remains an area of vigorous research with the promise of identifying a clinically viable inhibitor of efflux transporters for reversing TMZ resistance.

## 5. Approaches Employing Formulation for Improving the Neuropharmacokinetics of TMZ 

Multiple approaches of formulating TMZ into sustained release and targeted formulations have also been investigated. The rationale is to extend the half-life of TMZ by protecting its hydrolysis to MTIC, thereby enhancing its systemic exposure and subsequently its brain uptake. The elimination half-life of TMZ is pH-dependent [16,80]. At a physiological pH of 7.4, the half-life is 2 h, whereas in the acidic environment (pH < 4), it is about 24 h. This short elimination half-life in the blood is due to its conversion to MTIC, the pharmacologically active species that does not cross the BBB [15]. Due to its short-elimination half-life, TMZ is required to be given at higher doses with more frequent dosing [16,81]. Thus, reducing the rate of TMZ elimination by developing formulations that protect it from degradation in the blood and/or enhance its plasma–brain partitioning has the potential to maximize its therapeutic effects, while reducing its off-target distribution which often causes adverse effects [81]. Indeed, there is clinical evidence to suggest that using a lower dose of TMZ with protracted schedules is not only better tolerated, but may lead to the significant depletion of MGMT activity and better therapeutic outcomes [81]. Some of the important approaches for modified formulations of TMZ include the use of hydrogels, polymeric nanoparticles, inorganic nanoparticles, and liposomal delivery systems, as highlighted below in Table 5.

### 5.1. TMZ-Loaded Hydrogels

Hydrogel-based drug delivery systems employ a hydrophilic matrix, which incorporates hydrophilic compounds such as TMZ. Employing pre-clinical models, Adhikari et al. (2017) demonstrated that the amphiphilic diblock co-polypeptide hydrogels (DCHs) of 120-poly-lysine and 80-poly-leucine (K_80_L_120_) lead to the sustained delivery of TMZ. DCHs solidify at the human body temperature and provide a sustained release profile of the entrapped drug. DCHs with and without TMZ were found to be non-toxic to normal human astrocytes (NHAs) in vitro. In vivo studies showed that this delivery system enhanced the activity of TMZ in mice orthotopically implanted with GBM patient-derived xenografts. Treatment with TMZ-DCHs increased the survival from 28 days to 38 days when compared with TMZ treatment alone [82].

Zhao et al. (2019) focused on the codelivery of paclitaxel nanoparticles and TMZ through photopolymerizable hydrogels. This approach results in the synergistic inhibition of the colony formation of U87MG cells in vitro. The hydrogel coformulation of paclitaxel and TMZ was well-tolerated and suppressed tumor growth more efficiently than the single drugs in the U87MG orthotopic tumor model. Overall treatment with TMZ hydrogel as a single treatment increased survival in tumor-bearing mice by >35% when compared with untreated tumors. Treatment with a combination of paclitaxel nanoparticles and TMZ hydrogels further increased the survival to >50% [83].

### 5.2. TMZ-Loaded Nanoparticles

Fang et al. (2015) formulated a chitosan–biotin linked nanoparticle core of TMZ with a neutravidin-chlorotoxin shell. The nanoparticulate delivery system was observed to retain TMZ cytotoxicity in vitro and resulted in an increase in the BBB penetration of TMZ in mouse models [84]. In another study, Afzalipour et al. (2019) demonstrated the anti-glioma efficacy of TMZ-loaded triblock polymer-coated magnetic nanoparticles (MNPs) with a superparamagnetic iron oxide nanoparticle core, conjugated with folic acid. The use of folic acid for conjugation caused a marked increase in BBB permeability and tumor-specific distribution of TMZ-MNPs, leading to a significant decrease in tumor volume and the enhanced survival of rats bearing orthotopic glioma tumors [85].

Behrooz et al. (2022) developed B19 aptamer (Apt)-conjugated polyamidoamine (PAMAM) G4C12 dendrimer nanoparticles (Apt-NPs) to codeliver paclitaxel and TMZ to U87 GSCs. A substantial increase in the intracellular levels of both drugs in GSCs was observed by the investigators. Furthermore, the authors also noted that these Apt-NPs imparted significantly greater cytotoxic effects relative to the co-administration of the two given individually [86].

Another approach to enhance the delivery of TMZ to the GBM tumors is through the use of solid lipid nanoparticles (SLN). Ak et al. synthesized monocarboxylate transporter-1 targeted SLNs loaded with TMZ and BCNU. These SLNs were examined for their in vitro drug release profile and anti-apoptotic effects. The in vitro release profile demonstrated rapid drug release from the SLNs at the initial time, followed by a controlled continuous release. The SLN-formulated product was more effective in the induction of apoptosis relative to the treatment of cells with the combination without encapsulation [87].

Xu et al. (2020) developed nanostructured lipid carriers (NLCs) for the codelivery of TMZ and curcumin using the microemulsion technique. The NLCs were designed to impart a bi-phasic delivery with an initial burst release of curcumin followed by a sustained release of both drugs. The hypothesis was that the rapid release of curcumin would result in the pre-sensitization of GBMs prior to TMZ exposure. Furthermore, NLCs loaded with this combination showed synergistic anti-GBM effects in BALB/c nude mice implanted with subcutaneous C6 glioma tumors. The H&E-stained sections from the heart, liver, spleen, lung, kidney, and brain of the NLC-treated mice revealed no major changes relative to the placebo (saline), indicating minimal toxicological effects of the NLCs [88].

### 5.3. TMZ-Loaded Liposomal Delivery Systems

Liposomes are phospholipid structures, with a polar head and a lipophilic tail that closely resemble cellular membranes. The unique structure of liposomes, with a hydrophilic core and lipophilic phospholipid bilayer, enables the encapsulation of both hydrophilic and lipophilic molecules. This approach has been utilized to enhance the targeted delivery of TMZ to the brain, while reducing the peripheral degradation of TMZ. For instance, Gabay et al. (2021) developed the amyloid precursor peptide (APP)-linked liposomal carrier system for the targeted delivery of TMZ to the brain. The use of APP increased the BBB permeability of the liposomes containing TMZ both in vitro and in vivo (mouse model). Treatment with APP-linked liposomes containing TMZ resulted in delayed tumor growth and increased survival rates (45–70%) in tumor-bearing immunodeficient mice, when compared with the non-targeted TMZ liposomes and free TMZ [89].

In a 2018 study, Lam et al. investigated the use of transferrin to generate targeted liposomes for the codelivery of TMZ and the bromodomain inhibitor JQ1 to malignant brain tumors. Brain capillary endothelial cells express transferrin receptors that facilitate the transport of transferrin across the BBB. The levels of transferrin are elevated in malignant brain tumors, which can be exploited for targeted drug delivery. Consistent with this hypothesis, the researcher observed that treatment with TMZ-JQ1 transferrin linked liposomes enhanced the brain tumor localization of both drugs and increased survival rates in mouse xenograft models relative to single drug administration [90].

## 6. Conclusions

This article attempts to highlight some of the major mechanisms underlying resistance to TMZ in GBMs, as well as the current status of efforts to circumvent this obstacle. Unfortunately, TMZ remains one of the only drugs specifically approved for the treatment of GBMs. Intrinsic resistance, especially in GSCs, and acquired resistance after initiating drug therapy result in therapeutic failure. Although TMZ penetrates the BBB, GBM cells are no longer susceptible to TMZ upon the development of resistance, leading to tumor recurrence. Resistance to TMZ is multi-factorial and includes a combination of intracellular pharmacodynamic pathways (the enhanced neutralization of the DNA methylating effects of TMZ, improved DNA repair mechanisms, and the modulation of signaling pathways, leading to increased cell survival), and PK changes, such as the overexpression of efflux transporters that reduce intra-tumoral drug levels. Attempts to overcome TMZ resistance include the use of chemosensitizers that enhance the DNA-damaging impact of TMZ, approaches to alter systemic and CNS PK, such as hydrogel and nanoparticulate delivery systems, the concomitant use of P-gp inhibitors, and the co-formulation of TMZ and chemosensitizers. Most of the formulation approaches are still in pre-clinical stages, and efforts to accelerate their clinical investigation are warranted. Thus far, clinical investigations of chemosensitizers in combination with TMZ have recorded limited success in enhancing the PFS and OS rates of GBM patients. Likely confounding factors include GBM heterogeneity, multifactorial mechanisms of resistance to chemotherapy, the presence of GSCs carrying multiple mutations, and the limited BBB permeability of the investigational agents. Safety concerns with some of the compounds also restrict their potential to be used in combination with TMZ. However, optimistic prospects include the potential use of clinically approved drugs or investigational agents in the advanced clinical phases of development, such as PARP inhibitors, CDK4/6 inhibitors, proteosome inhibitors, and HDAC inhibitors with requisite PK properties, including brain penetration. Learnings from the ongoing studies will hopefully lead to optimized therapy that may include a combination of agents that can be used for patient-specific molecular targets.

## Figures and Tables

**Figure 1 ijms-25-03217-f001:**
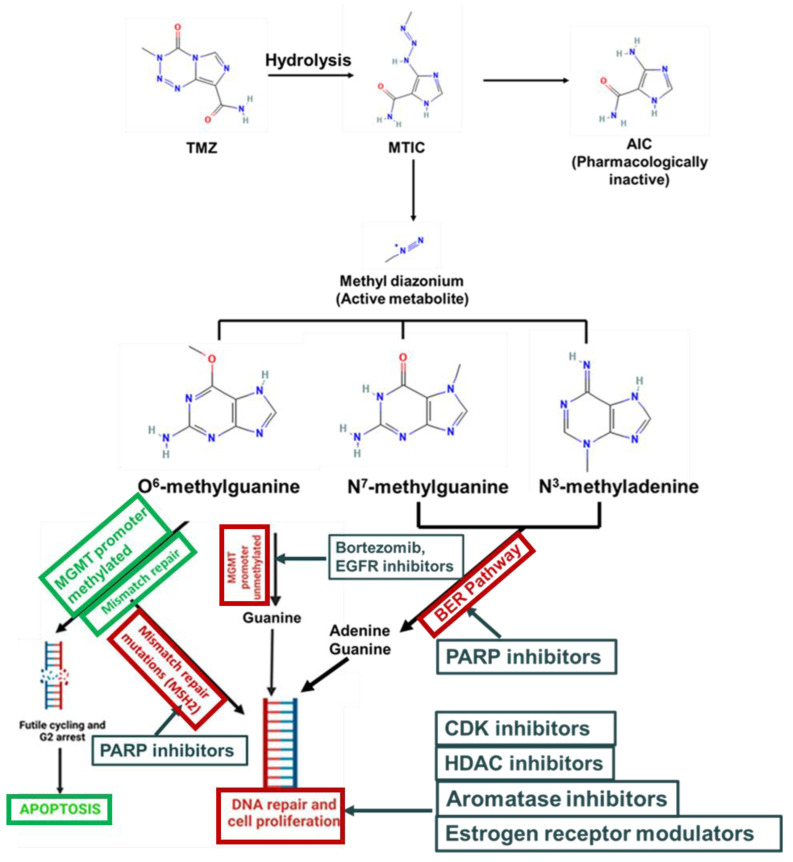
The TMZ mechanisms of action and intranuclear pathways for DNA repair leading to resistance and potential therapeutic approaches to overcome resistance to TMZ. At the physiological pH, TMZ is hydrolyzed to its active metabolite, 5-(3-Methyl-1-triazeno)imidazole-4-carboxamide(MTIC), which is subsequently converted to the pharmacologically active methyl diazonium and the inactive 5-aminoimidazole-4-carboxamide (AIC). The alkylating entity methyl diazonium participates in the methylation of O^6^ and N^7^ positions of the guanine and N^3^ position of adenine residues in DNA, respectively, leading to DNA double strand breaks (DSBs), which, in turn, activate apoptotic pathways in tumor cells [12,13]. Structures: 2D Structures were obtained from Pubchem (CID listed below): TMZ: 5394; MTIC: 76953; AIC: 9679; Methyl diazonium: 115287; O^6^-meG: 656275; N^7^-meG:135398679; N^3^-meA: 135398661.

**Table 1 ijms-25-03217-t001:** Histological subtypes of GBMs with a molecular signature (Wang et al., 2017) [4].

GBM Histological Subtype	Molecular Signature	Commonly Mutated Genes	Median OS
**Proneural**	PDGFRA, GABRB3, ERBB3, SOX10, HOXD3, HDAC2, EPHB1, CDKN1C	TP53, CDK4, OS9, PDGFRA, CDKN2B, EGFR	17.0
**Classical**	FGFR3, PTPRA, ELOVL2, SOX9, PAX6, CDH4, SEPT11, MEOX2	CDKN2B, EGFR	14.7
**Mesenchymal**	BCL3, TGFBI, ITGB1, LOX, VDR, IL6, COL1A2, MMP7	CDKN2B, NF1, NF-κB	11.5

**Table 2 ijms-25-03217-t002:** Pharmacokinetics and Pharmacodynamics of TMZ in GBM.

PK of TMZ in GBM Patients			
Parameter	Plasma	Brain ECF	References
C_max_ (μM)	28.3 (±16)	3.1 (±1.5)	[17]
T_max_ (h)	1.8 (±1.2)	2 (±0.8)	
t_1/2_ (h)	2.1 (±1.2)	2.9 (±1.6)	
AUC_(0-inf)_ (h·μg/mL)	17.1 (±6.8)	2.7 (±1.0)	
**Pharmacodynamics of TMZ in GBMs (In Vitro in Patient-Derived Cells)**			
**Parameter**	**TMZ-Sensitive GBMs**	**TMZ-Resistant GBMs**	**References**
IC_50_ (in vitro cytotoxicity)	1–50 μM	>100 μM	[18,19]

**Table 5 ijms-25-03217-t005:** Pre-clinical efforts to improve the PK characteristics of TMZ in GBMs through drug-delivery systems.

Drug Delivery System	Drug Combinations	PK Benefits	References
Targeted hydrogels	None	Sustained release; Reduced effects of degradation at physiological conditions; Increased elimination half-life; Potential reduction in toxicities	[82]
Nanoparticle incorporated hydrogels	Paclitaxel nanoparticles	Sustained release; Reduced effects of degradation at physiological conditions; Increased elimination half-life; Combination therapy using single formulation	[83]
Targeted nanoparticles	Chlorotoxin	Tumor targeted delivery; Sustained release; Reduced effects of degradation at physiological conditions; Increased elimination half-life; Potential reduction in toxicities	[84]
Magnetic nanoparticles	None	Sustained release; Increased elimination half-life	[85]
B19 aptamer conjugated nanoparticles	Paclitaxel	Sustained release; Increased intracellular retention	[86]
Solid lipid nanoparticles	BCNU	Sustained release; Targeted nanoparticles for improved penetration through the BBB	[87]
Nanostructured lipid carriers	Curcumin	Synergistic in vivo efficacy; Controlled release of TMZ; Minimal toxicities observed in mice	[88]
APP-linked Liposomes	None	Sustained release; Reduced peripheral degradation; Enhanced brain uptake	[89]
Transferrin-linked liposomes	JQ1 (Bromodomain inhibitor)	Sustained release; Reduced peripheral degradation; Enhanced brain and tumor uptake	[90]

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
