# Peer review of "A Review of Approaches to Potentiate the Activity of Temozolomide against Glioblastoma to Overcome Resistance"

_ijms, 2024, doi:10.3390/ijms25063217_

Round 1

Reviewer 1 Report

Comments and Suggestions for Authors

The manuscript “Potentiation of Temozolomide Activity in Glioblastoma Multiforme to Overcome Resistance: A Comprehensive Review” represents a well-written comprehensive literature review on the problems, the related molecular mechanisms and currently tested solutions, regarding the efficacy of temozolomide in the treatment of glioblastoma. Considering the dismal prognosis of glioblastoma, the topic undoubtedly is timely and important. Authors have summarized and explained the clinical shortcomings, that are associated with temozolomide, discussed the pathogenetic mechanisms of treatment failure and proposed solutions, that are realistic in nowadays medicine. As a detailed summary, the review will be useful for clinicians and scientists working in the given field.

The following question is of interest and thus should be highlighted in order to increase the scientific impact of the manuscript:

1) Authors have correctly noted that molecular features of glioblastoma (and of brain tumours in general) have an important role in classification. However, in diagnostic surgical pathology immunohistochemistry is frequently considered a fast and reliable substitute for molecular genetic tests therefore it would be valuable to refer shortly to immunohistochemistry-based molecular subtyping of glioblastoma and its association with response to treatment. See, please,

  • Molecular classification of diffuse gliomas. Jakovlevs A, et al. Pol J Pathol. 2019;70(4):246-258. DOI: 10.5114/pjp.2019.93126.
  • A simplified integrated molecular and immunohistochemistry-based algorithm allows high accuracy prediction of glioblastoma transcriptional subtypes. Orzan F, et al. Lab Invest. 2020;100(10):1330-1344. DOI: 10.1038/s41374-020-0437-0.

2) Few technical aspects could be improved:

2.1) In accordance with the “Instructions for authors”, Introduction remains a traditional part of review articles. Please, consider adding it.

2.2) In the text (page 2, line 50), Table 1 is noted in the context of the 4th edition (2007) of World Health Organisation (WHO) classification of CNS tumours. However, the title and contents of Table 1 reflect the newest version of classification, issued in 2021.

2.3) In the text, it has been noted that “morphological and molecular basis of classification of gliomas is detailed in Table 2” (page 2, line 78). However, the title and contents of Table 2 (page 3) are focused on treatment of glioblastoma, not classification.

2.4) Incidence rate should be expressed as number of cases per specified number of persons.

2.5) Please, define Lipinski’s rule of five.

2.6) In the line 133, authors refer to Table 4. However, the indicated data are present in Table 3. Please, solve the minor inaccuracies in numbering Tables 3 and 4 and referring to them in text.

2.7) Please, check lines 439 and 462 for minor misprints.

2.8) Check the formatting of references, please. It should be in a uniform style and follow the “Instructions of Authors”.

2.9) Information on authors’ contributions, funding and conflicts of interest is currently missing. Please, fill in these data in accordance with the “Instructions of Authors”; the explanations in these instructions are really helpful.

Finally, I would like to thank the authors for their contribution. It was a pleasure and a true honour to review this manuscript.

Comments on the Quality of English Language

The level of English is reasonably high. Few minor misprints should be corrected (see above, please). 

Author Response

We are grateful to this reviewer for this note underscoring the primary impetus for us to undertake writing this review article.

Reviewer 2 Report

Comments and Suggestions for Authors

Dear authors,

 You have done an excellent review about possible pharmacological ways to overcome resistance to temozolomide of in the treatment of glioblastoma. To my point of view the manuscript is too long. Review articles should have between 6,000 and 8,000 words (12 to 16 pages).

As the only concern is the treatment of glioblastoma, in order to shorten the text, the sections of classification of the CNS tumours and epidemiology of glioblastoma can be avoided.

The title must be corrected, because the term glioblastoma multiforme is not more in use. Instead the term of glioblastoma should be used.

Author Response

We thank the reviewer for her/his endorsement of our effort.   The word count of the original article was 9000.  With this suggestion to shorten the length of the manuscript, also expressed by the reviewer 3, our revision has reduced the word count to 7956.

Reviewer 3 Report

Comments and Suggestions for Authors

This paper is a bit amateurishly written. It is with sadness that I must recommend rejection as the paper now stands. The subject this paper purports to tackle - overview of some of the better known resistance pathways of GB to TMZ cytotoxicity - is important. Such a paper will be useful and welcome. I encourage the authors to write it. But this ms. is not such a paper.

TMZ resistance is a valuable subject for oncology researchers and clinicians to have summarized. 

This manuscript lacks proofreading by the neurooncology MD coauthors who obviously did not read or correct the submission. Grammar and English use are generally correct if too flowery. The scholarship behind this work was weak, but not weak enough to reject the paper. But combined with the disorganization and random nature of what resistance pathways are covered and what pathways are not covered plus the inclusion of extraneous, somewhat irrelevant, information, together are reasons to require a rewrite.

To the authors: Please take my comments with a collegial smile. My words are harsh but meant to help you become better communicators and better researchers. I would expect the same sharp criticism from you of any of my submissions. We all are on the same team trying to cure GB.

A suggestion for the rewrite: Authors, please make a simple list of mechanisms by which cytotoxicity of TMX becomes reduced. Then discuss each of these in a separate subheading - referencing data. Each subheading must directly connect to TMZ. The paper does not do this now. You are mixing growth driving pathways with TMZ resistance. Without specifically connecting the two [growth drive with TMZ effects or TMZ PKs] discussing growth drive becomes off topic.

A paper reviewing growth drive in GB would be a huge effort and probably go to hundreds of pages. Simply listing the growth drives identified in GB and a one paragraph overview with reference to reviews with a simplified diagram to the growth drive would be possible. But not in a paper on TMZ resistance unless, as above, the growth drive can be directly, mechanistically, and specifically connected to TMZ resistance.

There is a fair amount of empty verbiage that should be cut and can be cut with resultant increased reader ease and understanding. Authors- be succinct and direct. As an example in the Abstract simply list the resistance factors you discuss in your paper. Also please remove empty and self evident statements like “Thus, there is an urgent need to devise strategies to potentiate TMZ activity, to overcome drug resistance, and to reduce dose-dependent AEs”. What are you adding to cancer researchers and oncologists by saying this ? Nothing. Spare us.

Line 21 is error. One can say “can cause” but not “causes”. It would not be rare to see a pt cruise thru TMZ without major problems.

Line 24 is error. Grade 3 or 4 nausea is not common with TMZ. In long term TMZ treatment any nausea is not common and is easily treated.

This manuscript is not comprehensive. Many resistance mechanisms are not covered. Nor need they be if you specify “some of the currently recognized factors resulting in GB’s resistance to TMZ growth arrest”. 

The Introduction section should start at line 81. The verbiage in Introduction prior to that is unrelated to the subject of TMZ resistance. Classification of gliomas is way outside the subject the authors tackle. However, OK to start paper with a brief definition - characteristic features - of GB on H&E and outline molecular subtyping of GB.

Lines 83-83 are pointless. Say simply and directly what is important and relevant to your subject - TMZ resistance in GB. So start with line 85 “GB accounts for 14 % of all CNS tumors and 50 % of all malignant brain tumors. “ Note listing 14.2 % is silly. Do the authors really think the tenth place has meaning for this statistic ? It does not. Also note another indicator of no proofreading the authors stated “GBM accounts for 14.2% of all CNS tumors and 50.1% of all malignant tumors. “ This might lead to confusion by omitting “brain”.

The sloppy presentation and errors of Table 2 are emblematic of the amateurish presentation of the authors’ data. It must be rewritten by an oncologist who  actually has experience treating GB with TMZ.

Line 90 is deceptive and an example of overly flowery language. Don’t be cagey - list the % male/female without the preamble.

Line 106 error. Re “resection surgery” what other kind of resection is there other than surgery ?

Lines 110 to 112 are wrong. “Unfortunately, this combined modality of treatment is only marginally effective in altering the course of the disease, as the median survival of GBM patients is less than 8 months. Less than 6.8% of patients survive past 5 years after diagnosis [5]. “ Yes, this was what the authors’ reference 5 said but that is clearly in error. Any of the oncologists or neurosurgeons on the authors’ team would have seen and corrected this error had they read the paper before submitting it. Also I would ask the authors why they simply copy without digesting what ref 5 said. Proper medical writing usually requires quoting primary reports, not overarching broad general meta analysis like ref 5. Correct median OS examples: (differences below accounted for by economic constraints, cohort characteristics, and variations on Stupp used)

Ballo et al: median OS 17 months, 22 months with TTF

Guo et al median OS 12 months

Rozumenko et al  median OS 10 months

Mallick et al median OS 26 months

Trivedi et al median OS 23 months

etc.

Line 120 is deceptive. Reference the primary source of TMZ brain distribution determination and what the plasma:brain tissue ration is. [usually found to be 20% of plasma or a blood:brain ratio of 5:1 I think]. Most of us would not call that effectively penetrating the BBB since bone marrow problems are the main limiting factor to increasing TMZ dose or frequency. We ideally want the reverse ratio, 1:5, blood-to-brain. Few drugs have that.

Line 131, modern standard in medical writing is to avoid use of “respectively”. Altho this rule is often disobeyed, it shouldn’t be. Most readers must go back and correlate the two pairs of data.

Again attesting to a disregard for readers, the authors use “PK” without previously defining. Please reduce abbreviation use. Doing so will aid readers. On the other hand I don’t think AUC needs prior definition. Anyone interested in this work won’t need AUC defined. Re. use of BTB is nonstandard and you only use it 4 times so don’t abbreviate it. Blood-tumor partition is also not entirely relevant in that 90% of GB recurrences are within mormal appearing brain tissue that does not have overt tumor.  Even resection of an entire lobe does not cure GB.

On line 316 the use of BRCP is fine, OK to not define.

Line 315. Isn’t an alternate designation of P-gp usually ABCB1, not BCB1 ? 

I think the authors are in error re. Electrical resistance of brain capillaries. Amphiphiles generally cross the BBB better than strict lipophiles or hydrophiles. These amphiphiles are rather polar molecules as well as having areas of lipophilicity. Am I not correct in this ?

Line 340. Re elacridar, this is deceptive. There have been no published studies showing this in human GB [that I have seen].

Lines 345-350 must be rewritten. Several errors re. EGFR. There are several EGFR abnormalities common in GB. [Normal EGFR amplified, EGFRvIII amplified, EGFRvIII nonamplified, abnormal expression of EGFR agonists, abnormal proteolytic release of EGFR agonists, intracellular activation of EGFR without extracellular agonist binding, et al].

Why do the authors mention Wnt but not c-Met, Hedgehog, VEGFR, IL-8 receptors, IL-6 trceptors, Akt abnormalities, TNF receptors, IGF receptors, TGFbeta receptors, and dozens of other dysregulated signaling systems known to be hyperactive or malfunctioning in GB and contributing to TMZ resistance ?

Section 5 covers rather random aspects of GB growth vigor and TMZ resistance. The authors must directly connect any growth pathway they mention to data on how that growth pathway reduces TMZ effects.

Table 7 must list in separate column reference(s) to the listed item/drug.

Line 438 is example of no proofreading “apoptosis in GBM [47]The ERα antagonist “

Line 442. Error. Again, tamoxifen is not poorly tolerated usually.. No oncologist or GP would say tamoxifen is usually poorly tolerated.

The authors must quote and discuss Krajcer et al who tackles the same/similar subject as does the authors’ manuscript.

==================================

Ballo MT, Conlon P, Lavy-Shahaf G, Kinzel A, Vymazal J, Rulseh AM. Association of Tumor Treating Fields (TTFields) therapy with survival in newly diagnosed glioblastoma: a systematic review and meta-analysis. J Neurooncol. 2023 Jul 26. doi: 10.1007/s11060-023-04348-w. 

Guo X, Gu L, Li Y, Zheng Z, Chen W, Wang Y, Wang Y, Xing H, Shi Y, Liu D, Yang T, Xia Y, Li J, Wu J, Zhang K, Liang T, Wang H, Liu Q, Jin S, Qu T, Guo S, Li H, Wang Y, Ma W. Histological and molecular glioblastoma, IDH-wildtype: a real-world landscape using the 2021 WHO classification of central nervous system tumors. Front Oncol. 2023;13:1200815. doi: 10.3389/fonc.2023.1200815. 

Krajcer A, Grzywna E, Lewandowska-Łańcucka J. Strategies increasing the effectiveness of temozolomide at various levels of anti-GBL therapy. Biomed Pharmacother. 2023;165:115174. doi:10.1016/j.biopha.2023.115174. 

Mallick S, Gupta S, Amariyil A, Kunhiparambath H, Laviraj MA, Sharma S, Sagiraju HKR, Julka PK, Sharma D, Rath GK. Hypo-fractionated accelerated radiotherapy with concurrent and maintenance temozolomide in newly diagnosed glioblastoma: updated results from phase II HART-GBM trial. J Neurooncol. 2023 Jul 15. doi: 10.1007/s11060-023-04391-7. 

Rozumenko A, Kliuchka V, Rozumenko V, Daschakovskiy A, Fedorenko Z. Glioblastoma management in a lower middle-income country: Nationwide study of compliance with standard care protocols and survival outcomes in Ukraine. Neurooncol Pract. 2022;10(4):352-359. doi:10.1093/nop/npac094. 

Trivedi AG, Ramesh KK, Huang V, Mellon EA, Barker PB, Kleinberg LR, Weinberg BD, Shu HG, Shim H. Spectroscopic MRI-Based Biomarkers Predict Survival for Newly Diagnosed Glioblastoma in a Clinical Trial. Cancers (Basel). 2023;15(13):3524. doi:10.3390/cancers15133524. 

Comments on the Quality of English Language

This paper is a bit amateurishly written. It is with sadness that I must recommend rejection as the paper now stands. The subject this paper purports to tackle - overview of some of the better known resistance pathways of GB to TMZ cytotoxicity - is important. Such a paper will be useful and welcome. I encourage the authors to write it. But this ms. is not such a paper.

TMZ resistance is a valuable subject for oncology researchers and clinicians to have summarized. 

This manuscript lacks proofreading by the neurooncology MD coauthors who obviously did not read or correct the submission. Grammar and English use are generally correct if too flowery. The scholarship behind this work was weak, but not weak enough to reject the paper. But combined with the disorganization and random nature of what resistance pathways are covered and what pathways are not covered plus the inclusion of extraneous, somewhat irrelevant, information, together are reasons to require a rewrite.

To the authors: Please take my comments with a collegial smile. My words are harsh but meant to help you become better communicators and better researchers. I would expect the same sharp criticism from you of any of my submissions. We all are on the same team trying to cure GB.

A suggestion for the rewrite: Authors, please make a simple list of mechanisms by which cytotoxicity of TMX becomes reduced. Then discuss each of these in a separate subheading - referencing data. Each subheading must directly connect to TMZ. The paper does not do this now. You are mixing growth driving pathways with TMZ resistance. Without specifically connecting the two [growth drive with TMZ effects or TMZ PKs] discussing growth drive becomes off topic.

A paper reviewing growth drive in GB would be a huge effort and probably go to hundreds of pages. Simply listing the growth drives identified in GB and a one paragraph overview with reference to reviews with a simplified diagram to the growth drive would be possible. But not in a paper on TMZ resistance unless, as above, the growth drive can be directly, mechanistically, and specifically connected to TMZ resistance.

There is a fair amount of empty verbiage that should be cut and can be cut with resultant increased reader ease and understanding. Authors- be succinct and direct. As an example in the Abstract simply list the resistance factors you discuss in your paper. Also please remove empty and self evident statements like “Thus, there is an urgent need to devise strategies to potentiate TMZ activity, to overcome drug resistance, and to reduce dose-dependent AEs”. What are you adding to cancer researchers and oncologists by saying this ? Nothing. Spare us.

Line 21 is error. One can say “can cause” but not “causes”. It would not be rare to see a pt cruise thru TMZ without major problems.

Line 24 is error. Grade 3 or 4 nausea is not common with TMZ. In long term TMZ treatment any nausea is not common and is easily treated.

This manuscript is not comprehensive. Many resistance mechanisms are not covered. Nor need they be if you specify “some of the currently recognized factors resulting in GB’s resistance to TMZ growth arrest”. 

The Introduction section should start at line 81. The verbiage in Introduction prior to that is unrelated to the subject of TMZ resistance. Classification of gliomas is way outside the subject the authors tackle. However, OK to start paper with a brief definition - characteristic features - of GB on H&E and outline molecular subtyping of GB.

Lines 83-83 are pointless. Say simply and directly what is important and relevant to your subject - TMZ resistance in GB. So start with line 85 “GB accounts for 14 % of all CNS tumors and 50 % of all malignant brain tumors. “ Note listing 14.2 % is silly. Do the authors really think the tenth place has meaning for this statistic ? It does not. Also note another indicator of no proofreading the authors stated “GBM accounts for 14.2% of all CNS tumors and 50.1% of all malignant tumors. “ This might lead to confusion by omitting “brain”.

The sloppy presentation and errors of Table 2 are emblematic of the amateurish presentation of the authors’ data. It must be rewritten by an oncologist who  actually has experience treating GB with TMZ.

Line 90 is deceptive and an example of overly flowery language. Don’t be cagey - list the % male/female without the preamble.

Line 106 error. Re “resection surgery” what other kind of resection is there other than surgery ?

Lines 110 to 112 are wrong. “Unfortunately, this combined modality of treatment is only marginally effective in altering the course of the disease, as the median survival of GBM patients is less than 8 months. Less than 6.8% of patients survive past 5 years after diagnosis [5]. “ Yes, this was what the authors’ reference 5 said but that is clearly in error. Any of the oncologists or neurosurgeons on the authors’ team would have seen and corrected this error had they read the paper before submitting it. Also I would ask the authors why they simply copy without digesting what ref 5 said. Proper medical writing usually requires quoting primary reports, not overarching broad general meta analysis like ref 5. Correct median OS examples: (differences below accounted for by economic constraints, cohort characteristics, and variations on Stupp used)

Ballo et al: median OS 17 months, 22 months with TTF

Guo et al median OS 12 months

Rozumenko et al  median OS 10 months

Mallick et al median OS 26 months

Trivedi et al median OS 23 months

etc.

Line 120 is deceptive. Reference the primary source of TMZ brain distribution determination and what the plasma:brain tissue ration is. [usually found to be 20% of plasma or a blood:brain ratio of 5:1 I think]. Most of us would not call that effectively penetrating the BBB since bone marrow problems are the main limiting factor to increasing TMZ dose or frequency. We ideally want the reverse ratio, 1:5, blood-to-brain. Few drugs have that.

Line 131, modern standard in medical writing is to avoid use of “respectively”. Altho this rule is often disobeyed, it shouldn’t be. Most readers must go back and correlate the two pairs of data.

Again attesting to a disregard for readers, the authors use “PK” without previously defining. Please reduce abbreviation use. Doing so will aid readers. On the other hand I don’t think AUC needs prior definition. Anyone interested in this work won’t need AUC defined. Re. use of BTB is nonstandard and you only use it 4 times so don’t abbreviate it. Blood-tumor partition is also not entirely relevant in that 90% of GB recurrences are within mormal appearing brain tissue that does not have overt tumor.  Even resection of an entire lobe does not cure GB.

On line 316 the use of BRCP is fine, OK to not define.

Line 315. Isn’t an alternate designation of P-gp usually ABCB1, not BCB1 ? 

I think the authors are in error re. Electrical resistance of brain capillaries. Amphiphiles generally cross the BBB better than strict lipophiles or hydrophiles. These amphiphiles are rather polar molecules as well as having areas of lipophilicity. Am I not correct in this ?

Line 340. Re elacridar, this is deceptive. There have been no published studies showing this in human GB [that I have seen].

Lines 345-350 must be rewritten. Several errors re. EGFR. There are several EGFR abnormalities common in GB. [Normal EGFR amplified, EGFRvIII amplified, EGFRvIII nonamplified, abnormal expression of EGFR agonists, abnormal proteolytic release of EGFR agonists, intracellular activation of EGFR without extracellular agonist binding, et al].

Why do the authors mention Wnt but not c-Met, Hedgehog, VEGFR, IL-8 receptors, IL-6 trceptors, Akt abnormalities, TNF receptors, IGF receptors, TGFbeta receptors, and dozens of other dysregulated signaling systems known to be hyperactive or malfunctioning in GB and contributing to TMZ resistance ?

Section 5 covers rather random aspects of GB growth vigor and TMZ resistance. The authors must directly connect any growth pathway they mention to data on how that growth pathway reduces TMZ effects.

Table 7 must list in separate column reference(s) to the listed item/drug.

Line 438 is example of no proofreading “apoptosis in GBM [47]The ERα antagonist “

Line 442. Error. Again, tamoxifen is not poorly tolerated usually.. No oncologist or GP would say tamoxifen is usually poorly tolerated.

The authors must quote and discuss Krajcer et al who tackles the same/similar subject as does the authors’ manuscript.

==================================

Ballo MT, Conlon P, Lavy-Shahaf G, Kinzel A, Vymazal J, Rulseh AM. Association of Tumor Treating Fields (TTFields) therapy with survival in newly diagnosed glioblastoma: a systematic review and meta-analysis. J Neurooncol. 2023 Jul 26. doi: 10.1007/s11060-023-04348-w. 

Guo X, Gu L, Li Y, Zheng Z, Chen W, Wang Y, Wang Y, Xing H, Shi Y, Liu D, Yang T, Xia Y, Li J, Wu J, Zhang K, Liang T, Wang H, Liu Q, Jin S, Qu T, Guo S, Li H, Wang Y, Ma W. Histological and molecular glioblastoma, IDH-wildtype: a real-world landscape using the 2021 WHO classification of central nervous system tumors. Front Oncol. 2023;13:1200815. doi: 10.3389/fonc.2023.1200815. 

Krajcer A, Grzywna E, Lewandowska-Łańcucka J. Strategies increasing the effectiveness of temozolomide at various levels of anti-GBL therapy. Biomed Pharmacother. 2023;165:115174. doi:10.1016/j.biopha.2023.115174. 

Mallick S, Gupta S, Amariyil A, Kunhiparambath H, Laviraj MA, Sharma S, Sagiraju HKR, Julka PK, Sharma D, Rath GK. Hypo-fractionated accelerated radiotherapy with concurrent and maintenance temozolomide in newly diagnosed glioblastoma: updated results from phase II HART-GBM trial. J Neurooncol. 2023 Jul 15. doi: 10.1007/s11060-023-04391-7. 

Rozumenko A, Kliuchka V, Rozumenko V, Daschakovskiy A, Fedorenko Z. Glioblastoma management in a lower middle-income country: Nationwide study of compliance with standard care protocols and survival outcomes in Ukraine. Neurooncol Pract. 2022;10(4):352-359. doi:10.1093/nop/npac094. 

Trivedi AG, Ramesh KK, Huang V, Mellon EA, Barker PB, Kleinberg LR, Weinberg BD, Shu HG, Shim H. Spectroscopic MRI-Based Biomarkers Predict Survival for Newly Diagnosed Glioblastoma in a Clinical Trial. Cancers (Basel). 2023;15(13):3524. doi:10.3390/cancers15133524.

Author Response

First and foremost, we are grateful to this reviewer for taking the time to meticulously review our submission and to generously share her/his deep expertise.   As suggested by this reviewer, we have taken the critique, which is somewhat harsh, but often fair, as a constructive feedback.  Overall, we have shortened the manuscript, added new references to strengthen the information/data and corrected grammatical errors to the best of our ability. 

We do want to emphasize that the manuscript was reviewed by our clinical colleagues and a lot of important insights were added due to their participation.

Round 2

Reviewer 3 Report

Comments and Suggestions for Authors

The stated attempt of this paper was to review current pathways to 1] “potentiate activity of temozolomide against glioblastoma” and 2] “overcome resistance”. I cannot imagine how any paper short of a thick book could achieve this. There are many flaws and problems with this paper, below are just a few examples. 

The authors have invested much work in this paper and I want to be more positive about this paper but I cannot see it benefiting anyone the way it is currently written and the way it is currently constructed. It is not a review of paths to overcome resistance to temozolomide. Many other paths are not discussed but are in the peer-reviewed literature. The authors have chosen to discuss only some of the currently popular pathways. Also resistance pathways are related but conceptually different from pathways to augment a drug. The authors attempt to do both and fail on both accounts. They omit many peer reviewed pathways in each category. A similar capricious choosing of what to discuss and what not repeats throughout the paper.

The authors discuss pathways that were explored [on sound preclinical data and reasoning] but failed to benefit, failed to augment temozolomide in phase 3 and 4 studies. How does that fit under their stated goal ? It doesn' t.

Line 148. Error in reference notation. 

Table 3 is quite useful. Table 4 would be useful if references were from primary sources. But they weren't. Reference 10 is unacceptable. Anyone with experience using temozolomide will know that lymphocyte reductions are more common than neutrophil reductions. Authors must cite primary research data in table 4 not the p.i.

Table 6 is grossly deceptive and incorrect. It would be correct if the title to it was “Investigational status of selected compounds assessed for potentiation of TMZ activity…”.  But then the authors would need to state why they selected these pathways and not others. Also Table 6 leaves out dozens of other drugs being explored as inhibitors of their listed target, and leaves out dozens of rational targets being explored.

How can the authors rewrite their work such that it would be useful to glioblastoma or oncology researchers and/or clinicians ? A first step must be limiting their scope and then be thorough. I see the tremendous amount of work the team put into writing this paper but it will not be of much use to anyone as it now stands.

Lines 619 to 629 is just one example problem among many in this paper. First, GB stem cell markers are also functional mediators. Discussing that, marker = mediator, would be a useful paper, to collect all past literature on markers of GB stem subpopulation, then discuss the data we know on how each marker actually mediates one or another function that creates or contributes to one or more of the attributes of GB cell stemness. This would have to include discussing that the category of GB cell “stemness” is not a unitary entity as a range of attributes. The glass of water on my table is intact, it is shattered in pieces after falling on the floor. Intact or shattered. Stem-nonstem are not analogous to that. There is nonuniformity of GB cell stem markers and range of attributes to which we give the name GB stem cell. All the cells within that group we call GB stem cells do not share all the attributes by which we define the group GB stem cell.

There are a dozen other glossed over, vague, deceptive paragraphs as the above example.

Fig. 2 again must be titled  “Selective mechanisms of resistance to TMZ and targeted therapeutics to overcome…” Dozens of drugs, dozens of biochemistry pathways are not shown. Also the caveat "simplified" must be added to the figure title.

Section 3.4 has numerous errors. Many amphiphiles attain 10-20 times higher brain tissue levels than plasma levels. And as the authors point out the BBB is not a limiting factor in temozolomide effectiveness - even 1000 microM temozolomide doesn’ t kill the resistant subpopulation.

Re Table 7 and its discussion is unclear. Why discuss PK if PK is not the limiting factor in temozolomide resistance ? 

Section 4.2 is another example of verbiage that has no place in an academic article. As the authors stated, autophagy has a research database showing both aiding GB cell survival and an equally compelling database on hindering GB cell survival. We are no wiser reading their 4.2 paragraphs. It would indeed be a worthwhile paper if the authors could write a paper resolving these seemingly contradictory datasets. Also the random nature of reporting on chloroquine as autophagy inhibitor. Why omit mention of the dozen other clinically used autophagy inhibitors that have been suggested or tried in preclinical GB models ?

Re 4.5, again the authors do not discuss paradox of zero clinical benefit of EGFR inhibitors yet EGFR inhibitors had robust data that these should help and did work in preclinical models. In fact we do know several reasons why this discrepancy exists. That it self would be a worthy paper.

Re section 4.6 re Wnt is an important subject in GB research. The dozens of papers reporting dozens of ways we have attempted to inhibit its growth promoting functions in GB are simply ignored. Why ? A review of Wnt signaling in GB and the many drugs to inhibit Wnt that are being explored are worthy of an entire paper, and that paper would be of the same length as the authors’ current paper.

Comments on the Quality of English Language

The stated attempt of this paper was to review current pathways to 1] “potentiate activity of temozolomide against glioblastoma” and 2] “overcome resistance”. I cannot imagine how any paper short of a thick book could achieve this. There are many flaws and problems with this paper, below are just a few examples. 

The authors have invested much work in this paper and I want to be more positive about this paper but I cannot see it benefiting anyone the way it is currently written and the way it is currently constructed. It is not a review of paths to overcome resistance to temozolomide. Many other paths are not discussed but are in the peer-reviewed literature. The authors have chosen to discuss only some of the currently popular pathways. Also resistance pathways are related but conceptually different from pathways to augment a drug. The authors attempt to do both and fail on both accounts. They omit many peer reviewed pathways in each category. A similar capricious choosing of what to discuss and what not repeats throughout the paper.

The authors discuss pathways that were explored [on sound preclinical data and reasoning] but failed to benefit, failed to augment temozolomide in phase 3 and 4 studies. How does that fit under their stated goal ? It doesn' t.

Line 148. Error in reference notation. 

Table 3 is quite useful. Table 4 would be useful if references were from primary sources. But they weren't. Reference 10 is unacceptable. Anyone with experience using temozolomide will know that lymphocyte reductions are more common than neutrophil reductions. Authors must cite primary research data in table 4 not the p.i.

Table 6 is grossly deceptive and incorrect. It would be correct if the title to it was “Investigational status of selected compounds assessed for potentiation of TMZ activity…”.  But then the authors would need to state why they selected these pathways and not others. Also Table 6 leaves out dozens of other drugs being explored as inhibitors of their listed target, and leaves out dozens of rational targets being explored.

How can the authors rewrite their work such that it would be useful to glioblastoma or oncology researchers and/or clinicians ? A first step must be limiting their scope and then be thorough. I see the tremendous amount of work the team put into writing this paper but it will not be of much use to anyone as it now stands.

Lines 619 to 629 is just one example problem among many in this paper. First, GB stem cell markers are also functional mediators. Discussing that, marker = mediator, would be a useful paper, to collect all past literature on markers of GB stem subpopulation, then discuss the data we know on how each marker actually mediates one or another function that creates or contributes to one or more of the attributes of GB cell stemness. This would have to include discussing that the category of GB cell “stemness” is not a unitary entity as a range of attributes. The glass of water on my table is intact, it is shattered in pieces after falling on the floor. Intact or shattered. Stem-nonstem are not analogous to that. There is nonuniformity of GB cell stem markers and range of attributes to which we give the name GB stem cell. All the cells within that group we call GB stem cells do not share all the attributes by which we define the group GB stem cell.

There are a dozen other glossed over, vague, deceptive paragraphs as the above example.

Fig. 2 again must be titled  “Selective mechanisms of resistance to TMZ and targeted therapeutics to overcome…” Dozens of drugs, dozens of biochemistry pathways are not shown. Also the caveat "simplified" must be added to the figure title.

Section 3.4 has numerous errors. Many amphiphiles attain 10-20 times higher brain tissue levels than plasma levels. And as the authors point out the BBB is not a limiting factor in temozolomide effectiveness - even 1000 microM temozolomide doesn’ t kill the resistant subpopulation.

Re Table 7 and its discussion is unclear. Why discuss PK if PK is not the limiting factor in temozolomide resistance ? 

Section 4.2 is another example of verbiage that has no place in an academic article. As the authors stated, autophagy has a research database showing both aiding GB cell survival and an equally compelling database on hindering GB cell survival. We are no wiser reading their 4.2 paragraphs. It would indeed be a worthwhile paper if the authors could write a paper resolving these seemingly contradictory datasets. Also the random nature of reporting on chloroquine as autophagy inhibitor. Why omit mention of the dozen other clinically used autophagy inhibitors that have been suggested or tried in preclinical GB models ?

Re 4.5, again the authors do not discuss paradox of zero clinical benefit of EGFR inhibitors yet EGFR inhibitors had robust data that these should help and did work in preclinical models. In fact we do know several reasons why this discrepancy exists. That it self would be a worthy paper.

Re section 4.6 re Wnt is an important subject in GB research. The dozens of papers reporting dozens of ways we have attempted to inhibit its growth promoting functions in GB are simply ignored. Why ? A review of Wnt signaling in GB and the many drugs to inhibit Wnt that are being explored are worthy of an entire paper, and that paper would be of the same length as the authors’ current paper.

Author Response

The stated attempt of this paper was to review current pathways to 1] “potentiate activity of temozolomide against glioblastoma” and 2] “overcome resistance”. I cannot imagine how any paper short of a thick book could achieve this. There are many flaws and problems with this paper, below are just a few examples. 

We disagree.   There are other review articles written on this and other topics.  Not every review article can delve into the depth of every possible mechanism and publication.   There has to be difference between a review article and a thick book.

The authors have invested much work in this paper and I want to be more positive about this paper but I cannot see it benefiting anyone the way it is currently written and the way it is currently constructed. It is not a review of paths to overcome resistance to temozolomide. Many other paths are not discussed but are in the peer-reviewed literature. The authors have chosen to discuss only some of the currently popular pathways. Also resistance pathways are related but conceptually different from pathways to augment a drug. The authors attempt to do both and fail on both accounts. They omit many peer reviewed pathways in each category. A similar capricious choosing of what to discuss and what not repeats throughout the paper.

Again, this review article focused on major pathways of temozolomide resistance.  Due to scope and space constraints, we cannot include every pre-clinical and clinical effort undertaken to over resistance to temozolomide.   Taking this reviewer’s suggestion, however, we provide a more direct mechanistic connection to the investigated chemosensitization strategy.   We believe that connecting those dots have certainly further strengthened the manuscript. 

The authors discuss pathways that were explored [on sound preclinical data and reasoning] but failed to benefit, failed to augment temozolomide in phase 3 and 4 studies. How does that fit under their stated goal ? It doesn't.

In neuro-oncology research we are all painfully aware that the therapeutic approaches have not been successful.  Just because the clinical success eludes us, it does mean important lessons cannot be learnt for the future.  That is precisely the rationale for this review article and similar articles.  Where clear or possible, we have provided the reasons for the failure of chemosensitization of the indicated efforts

Line 148. Error in reference notation.-------------------

We do not see error in reference notation on Line 148. 

.   

Table 3 is quite useful. Table 4 would be useful if references were from primary sources. But they weren't. Reference 10 is unacceptable. Anyone with experience using temozolomide will know that lymphocyte reductions are more common than neutrophil reductions. Authors must cite primary research data in table 4 not the p.i.

We have modified the Table 4 as suggested by citing the primary research. 

Table 6 is grossly deceptive and incorrect. It would be correct if the title to it was “Investigational status of selected compounds assessed for potentiation of TMZ activity…”.  But then the authors would need to state why they selected these pathways and not others. Also Table 6 leaves out dozens of other drugs being explored as inhibitors of their listed target, and leaves out dozens of rational targets being explored.

We have modified the title as suggested.  Our attempt here is to focus on the major pathways contributing to temozolomide resistance and chemosensitization efforts undertaken to target those mechanisms.  As indicated above, it is not practical to list every mechanism and each investigational agent used.  

How can the authors rewrite their work such that it would be useful to glioblastoma or oncology researchers and/or clinicians ? A first step must be limiting their scope and then be thorough. I see the tremendous amount of work the team put into writing this paper but it will not be of much use to anyone as it now stands.

Yes, we have indicated the limited scope and also substantially revised the manuscript.

Lines 619 to 629 is just one example problem among many in this paper. First, GB stem cell markers are also functional mediators. Discussing that, marker = mediator, would be a useful paper, to collect all past literature on markers of GB stem subpopulation, then discuss the data we know on how each marker actually mediates one or another function that creates or contributes to one or more of the attributes of GB cell stemness. This would have to include discussing that the category of GB cell “stemness” is not a unitary entity as a range of attributes. The glass of water on my table is intact, it is shattered in pieces after falling on the floor. Intact or shattered. Stem-nonstem are not analogous to that. There is nonuniformity of GB cell stem markers and range of attributes to which we give the name GB stem cell. All the cells within that group we call GB stem cells do not share all the attributes by which we define the group GB stem cell.

The point is well made.  However, we had already indicated the extensive heterogeneity within GBM tumor mass, particularly in the stem cell population.    We have further clarified it in the revised manuscript.

There are a dozen other glossed over, vague, deceptive paragraphs as the above example.

While we disagree with this characterization, we have substantially revised the write-up to provide clarity and additional details as deemed fit.    

Fig. 2 again must be titled  “Selective mechanisms of resistance to TMZ and targeted therapeutics to overcome…” Dozens of drugs, dozens of biochemistry pathways are not shown. Also the caveat "simplified" must be added to the figure title.

The Figure 2 title has been changed.   it is impractical to list every mechanism and each investigational agent used.  

Section 3.4 has numerous errors. Many amphiphiles attain 10-20 times higher brain tissue levels than plasma levels. And as the authors point out the BBB is not a limiting factor in temozolomide effectiveness - even 1000 microM temozolomide doesn’ t kill the resistant subpopulation.

We addressed the comment about amphiphiles raised by this reviewer in the previous version.  We are in disagreement with her/him about that.   The BBB itself may not be stumbling block for temozolomide but its short elimination half-life due to its conversion to MTIC at physiological pH restricts the amount of drug available in plasma to be transported across the BBB.   We have clarified this point.   Yes, once the tumor acquires resistance it is difficult to deliver effective levels of temozolomide.  But the point is that improving the drug availability into the tumor mass may just prevent GBM cells from acquiring resistance.  Also, the conversation about BBB as a limitation is applicable not just to temozolomide but also to compounds being investigated as chemosensitizers.

Re Table 7 and its discussion is unclear. Why discuss PK if PK is not the limiting factor in temozolomide resistance ? 

As indicated above, the pH-dependent elimination half-life of temozolomide and its potential impact on temozolomide availability to the tumor mass is a limiting factor.  We have elaborated that in the revised manuscript.

Section 4.2 is another example of verbiage that has no place in an academic article. As the authors stated, autophagy has a research database showing both aiding GB cell survival and an equally compelling database on hindering GB cell survival. We are no wiser reading their 4.2 paragraphs. It would indeed be a worthwhile paper if the authors could write a paper resolving these seemingly contradictory datasets. Also the random nature of reporting on chloroquine as autophagy inhibitor. Why omit mention of the dozen other clinically used autophagy inhibitors that have been suggested or tried in preclinical GB models ?

Addressing the controverted issue of autophagy relevant to GBM therapy is not feasible here.  However, a few compounds that have investigated for potentiating temozolomide activity have been included.  This can prompt an interested reader, not all of whom are necessarily as well versed in GBM biology, as this reviewer. 

Re 4.5, again the authors do not discuss paradox of zero clinical benefit of EGFR inhibitors yet EGFR inhibitors had robust data that these should help and did work in preclinical models. In fact we do know several reasons why this discrepancy exists. That it self would be a worthy paper.

The reasons for promising EGFR inhibitors failing in the clinical setting are briefly discussed.   The above-indicated paradox by the reviewer is beyond the scope of this article.

Re section 4.6 re Wnt is an important subject in GB research. The dozens of papers reporting dozens of ways we have attempted to inhibit its growth promoting functions in GB are simply ignored. Why ? A review of Wnt signaling in GB and the many drugs to inhibit Wnt that are being explored are worthy of an entire paper, and that paper would be of the same length as the authors’ current paper.

We have focused on the role of Wnt signaling as relevant to temozolomide resistance.   More detailed discussion merits another manuscript, as suggested by the reviewer.

Round 3

Reviewer 3 Report

Comments and Suggestions for Authors

The subject the authors tackle is important. Their paper would be a useful overview if the authors could tighten their presentation and limit the scope of their work to what the title promises. If they cannot tighten the manuscript this work should not be published. The desultory, unfocused, capricious side excursions must be eliminated.

The paper seems to lack input from a clinician with experience treating GB. The ms. would benefit from input from well seasoned neuro-oncologist who could modify elements that come out of a package insert but are not commonly seen clinically. 

Lines 15 to 26 are empty verbiage. Although this kind of introduction is not rare, I view it as wasteful of readers’ time. Use the Abstract to convey as much of the substance of your article as possible in 250 words. So simply saying “GB is nearly invariably fatal with current standard treatment of resection, temozolomide and irradiation.” That better treatments are needed is obvious. Then jump right into what some recent approaches to augment TMZ effects have been. Again the title is deceptive or wrong. Many recent published efforts to augment TMZ were not mentioned by the authors. They only mentioned popular, fashionable ones.

Section “1.1. Overview of Glioblastoma”, should be drastically shortened to a few simple lines. The content of this section is not relevant to the subject of “Approaches to Potentiate Activity of Temozolomide Against Glioblastoma…” Tables 1 and 2 are irrelevant to the authors’ subject as well. Section 1.1.1. Epidemiology of GBM, is irrelevant. Section 1.1.2. can be much shortened.

Section 1.2 is well done and relevant. Table 3 is nicely done and helpful. The text preceding Section 1.2 should not be more than 20-30 lines or so.  Figure 1 is too large but is helpful. All abbreviations in Fig 1 must be explained in legend.

I would eliminate Table 4 and simply state in text “neutropenia (5-8%), thrombocytopenia (4-11%), fatigue (9%) and headache (4%) are the common grade 3-4 adverse effects of TMZ.”

Table 5 has inconsistencies. Under references just list numbers. “outcomes” and “limitations” can be shortened and combined to benefit readers. 

Line 258. Re. use of “TTF” I advocate stating longhand, TTF refers to an external device that exposes the area of GB resection to an ordinary radiofrequency (150 kHz) electromagnetic field. or something along those lines. Using term TTF without specifying what fields implies a mysterious or special field. I know many colleagues think of the Optune TTF device as something special or a bit mysterious. It isnt. 

Lines 425 to 433. Is an example, emblematic of the problem with this paper. I don’t see how mentioning Wnt-beta-catenin facilitates GB growth and treatment resistance will be of use to anyone. If an article is going to help us it must review all the Wnt signaling pathways and the dozen currently marketed drugs inhibit one or another of these pathways and how we have been trying to block this element of GB growth and TMZ resistance. The authors, in their response to my previous review of their manuscript correctly comment that to address each and every TMZ resistance pathway would be beyond the scope of their paper. I agree but then what use is this paper ? Just listing some resistance pathways doesnt help us. Also there are a dozen growth driving TMZ resistance pathways the authors ignore. Why did they pick these and not others ?

Line 461 on page 12 starts addressing the stated object of their review. I dont see the relevance of much of the verbiage before this. All the verbiage prior to line 461 can and should be put in maximum 2 pages of manuscript.

Re Table 6, some of the references the authors list have shown clear failure to improve TMZ effects. Again simple listing drug class and its “TMZ resistance pathway targeted” is a disservice to readers. The listed drugs have many more, and more important, GB inhibiting effects than those listed.

Re. section 4, again I dont see the point of the authors cursory and haphazard mentioning a few elements of each of these elements. Pick any one of them and really review it so we come to a thorough understanding of that pathway and efforts to overcome it and why.how those efforts have failed so far. It is ok to simply state the unresolved paradox too. For example we have data that autophagy enhances TMZ and data showing autophagy is a TMZ resistance factor. Or the discrepant EGFR data. 

I urge the authors to pick one and do it justice:

4.1. DNA-damage repair-targeting drugs

4.2. Autophagy modulators

4.3. Estrogen receptor modulators

4.4. Aromatase Inhibitors

4.5. EGFR inhibitors

4.6. Wnt/β-catenin pathway inhibitors

4.7. Histone Deacetylase (HDAC) inhibitors

4.8. Cell cycle checkpoint inhibitors

4.9. JAK/STAT pathway inhibitors

4.10. Inhibitors of multidrug transporting proteins

Hopping over each as the authors did, with capricious, selected elements, ignoring many related elements of each does not help us.

Section 5. “Approaches for improving the neuropharmacokinetics of TMZ”  is useful and properly presented but mislabeled. The authors review systemic and neuro pharmacokinetics. Again multiple efforts to improve TMZ pharmacokinetics were not mentioned. Why the authors chose the elements they chose to discuss and why they omitted other efforts was unclear.

I dont think many of my colleagues or patients would agree with “The clinical use of TMZ is fraught with numerous debilitating AEs that significantly impact the QoL of GBM patients.“ The authors dont have to overstate their case. We all agree TMZ has severe limitations. Mainly that people are usually dead within a few years of diagnosis and TMZ doesnt change that gruesome fact much.

Comments on the Quality of English Language

English language use is adequate.

Author Response

Please see our point-by-point responses as highlighted below:

The paper seems to lack input from a clinician with experience treating GB. The ms. would benefit from input from well seasoned neuro-oncologist who could modify elements that come out of a package insert but are not commonly seen clinically. 

Perhaps the reviewer 3 missed it but two of our co-authors who are clinical experts, Drs. Soma Sengupta (neuro-oncologist) and Trisha Wise-Draper (Hematologist/Oncologist) have reviewed and made substantial contribution to the write up of this manuscript.   Since even top-notch professionals can sometimes disagree, we sought input form a neuro-oncologist, Dr. Lalanthica Yogendran, who has meticulously reviewed and edited the manuscript in response to this reviewer.  She is now listed as a co-author.

Lines 15 to 26 are empty verbiage. Although this kind of introduction is not rare, I view it as wasteful of readers’ time. Use the Abstract to convey as much of the substance of your article as possible in 250 words. So simply saying “GB is nearly invariably fatal with current standard treatment of resection, temozolomide and irradiation.” That better treatments are needed is obvious. Then jump right into what some recent approaches to augment TMZ effects have been. Again the title is deceptive or wrong. Many recent published efforts to augment TMZ were not mentioned by the authors. They only mentioned popular, fashionable ones.

The abstract has been shortened and is now about 210 words.   We do not agree that the title is deceptive.  The attempt has been to include the efforts that impact major, clinically relevant pathways that result in resistance to temozolomide.  To say that we cherry-picked the “popular, fashionable ones” is misleading.

Section “1.1. Overview of Glioblastoma”, should be drastically shortened to a few simple lines. The content of this section is not relevant to the subject of “Approaches to Potentiate Activity of Temozolomide Against Glioblastoma…” Tables 1 and 2 are irrelevant to the authors’ subject as well. Section 1.1.1. Epidemiology of GBM, is irrelevant. Section 1.1.2. can be much shortened.

We are in disagreement for several reasons. 

  • Not everyone reading this paper is likely to be an expert in neuro-oncology. Review articles greatly help readers from other areas of research and who are not steeped into this field.   Thus, setting up an adequate context is important.
  • We included epidemiological data based on first round of review by one of the reviewers; however, we have shortened the write up and more importantly, updated the incidence data for 2023.
  • Reviewer 3 has previously suggested that there is considerable variability in the overall survival. Including a few molecular details and outcomes for histological sub-types of GBM provides important insights.
  • We have considerably shortened the introduction part to just include the details for GBM. (Lines 38-45)

Section 1.2 is well done and relevant. Table 3 is nicely done and helpful. The text preceding Section 1.2 should not be more than 20-30 lines or so.  Figure 1 is too large but is helpful. All abbreviations in Fig 1 must be explained in legend.

Thanks.

I would eliminate Table 4 and simply state in text “neutropenia (5-8%), thrombocytopenia (4-11%), fatigue (9%) and headache (4%) are the common grade 3-4 adverse effects of TMZ.”

We have incorporated this change.

Table 5 has inconsistencies. Under references just list numbers. “outcomes” and “limitations” can be shortened and combined to benefit readers. 

We have incorporated this change.

Line 258. Re. use of “TTF” I advocate stating longhand, TTF refers to an external device that exposes the area of GB resection to an ordinary radiofrequency (150 kHz) electromagnetic field. or something along those lines. Using term TTF without specifying what fields implies a mysterious or special field. I know many colleagues think of the Optune TTF device as something special or a bit mysterious. It isnt. 

We have incorporated this change (Lines 248-252). 

Lines 425 to 433. Is an example, emblematic of the problem with this paper. I don’t see how mentioning Wnt-beta-catenin facilitates GB growth and treatment resistance will be of use to anyone. If an article is going to help us it must review all the Wnt signaling pathways and the dozen currently marketed drugs inhibit one or another of these pathways and how we have been trying to block this element of GB growth and TMZ resistance. The authors, in their response to my previous review of their manuscript correctly comment that to address each and every TMZ resistance pathway would be beyond the scope of their paper. I agree but then what use is this paper ? Just listing some resistance pathways doesnt help us. Also there are a dozen growth driving TMZ resistance pathways the authors ignore. Why did they pick these and not others ?

We have added the pathways of resistance directly linked with TMZ and removed the pathways targeting GBM growth drive. (Section 3.4.1 EGFR variants and 3.4.2 Wnt signaling pathways). 

Line 461 on page 12 starts addressing the stated object of their review. I dont see the relevance of much of the verbiage before this. All the verbiage prior to line 461 can and should be put in maximum 2 pages of manuscript.

We have considerably shortened the introduction section and pharmacology of TMZ.

Re Table 6, some of the references the authors list have shown clear failure to improve TMZ effects. Again simple listing drug class and its “TMZ resistance pathway targeted” is a disservice to readers. The listed drugs have many more, and more important, GB inhibiting effects than those listed.

 We have modified this table to include the drugs directly affecting resistance to TMZ.

Re. section 4, again I dont see the point of the authors cursory and haphazard mentioning a few elements of each of these elements. Pick any one of them and really review it so we come to a thorough understanding of that pathway and efforts to overcome it and why.how those efforts have failed so far. It is ok to simply state the unresolved paradox too. For example we have data that autophagy enhances TMZ and data showing autophagy is a TMZ resistance factor. Or the discrepant EGFR data. 

I urge the authors to pick one and do it justice:

4.1. DNA-damage repair-targeting drugs

4.2. Autophagy modulators

4.3. Estrogen receptor modulators

4.4. Aromatase Inhibitors

4.5. EGFR inhibitors

4.6. Wnt/β-catenin pathway inhibitors

4.7. Histone Deacetylase (HDAC) inhibitors

4.8. Cell cycle checkpoint inhibitors

4.9. JAK/STAT pathway inhibitors

4.10. Inhibitors of multidrug transporting proteins

Hopping over each as the authors did, with capricious, selected elements, ignoring many related elements of each does not help us.

There is nothing capricious about it.  We attempted a broad review and did not focus on promises and pitfalls associated with just one pathway.   Again, we did not cherry pick the pathways but listed only those that impact TMZ activity, directly or indirectly.  But we agree that some of the content can be eliminated for brevity. We have removed the autophagy modulators and JAK/STAT pathway inhibitors from the manuscript and included inhibitors of pathways directly linked with resistance to TMZ.

Section 5. “Approaches for improving the neuropharmacokinetics of TMZ”  is useful and properly presented but mislabeled. The authors review systemic and neuro pharmacokinetics. Again multiple efforts to improve TMZ pharmacokinetics were not mentioned. Why the authors chose the elements they chose to discuss and why they omitted other efforts was unclear.

We have included a few more examples of the approaches for improved temozolomide delivery.  This again would be an exhaustive list and we have provided useful examples of different approaches such as the use of hydrogel, nanoparticles, liposomal delivery etc.
